# A Study on the Effects of Digital Learning Sheet Design Strategy on the Learning Motivation and Learning Outcomes of Museum Exhibition Visitors

Tien-Li Chen [1], Yun-Chi Lee [2,*] and Chi-Sen Hung [3]

1 Department of Industrial Design, College of Design, National Taipei University of Technology, Taipei 10608, Taiwan; chentl@mail.ntut.edu.tw
2 Doctoral Program in Design, College of Design, National Taipei University of Technology, Taipei 10608, Taiwan
3 Department of Communication Design, National Taichung University of Science and Technology, Taichung 404336, Taiwan; s6317666@gmail.com
* Correspondence: yunchilee0604@gmail.com; Tel.: +886-933-561-741

**Abstract:** This study focuses on "digital learning sheets" by exploring the effects of different design strategies of the digital learning sheet on visitors' motivation and learning outcomes. This study chose the woodcraft themed exhibition as a case study, adopting the learning sheet design principles proposed by Hooper-Greenhill in order to design three types of digital learning sheets for this exhibition. A control group of students who did not use the sheets and three experimental groups of students who used the sheets were invited to visit the exhibition for the purpose of examining the impact of different strategies of digital learning sheet design on the "learning motivation" and "learning outcomes" of the visitors. The study results show that among the four learning motivations of Attention, Relevance, Confidence, and Satisfaction, the digital learning sheet designed with the "principle of sensory exploration of physical objects" had the highest effectiveness among the various design strategies. In terms of three aspects of learning outcomes: Cognitive, Affective, and Psychomotor, the three types of digital learning design strategies do not produce significant differences in the affective impact on children. As for Cognitive and Psychomotor, students learn best when they use digital learning sheets designed with the "design principle of sensory and exploration of physical objects". The results of this study will provide future exhibition planners, digital learning designers, and educators with precise and practical references.

**Keywords:** digital learning sheet design; museum exhibition planning; informal education; learning motivation; learning outcomes

## 1. Foreword

A city that is constantly moving forward and improving towards a better vision requires a sound education infrastructure as well as citizens with a lifelong learning mindset. The development of learning cities is related to the issues of sustainable development and global citizenship [1]. According to the United Nations Educational, Scientific and Cultural Organization (UNESCO), holistic learning strengthens citizenship, social integration, economic development, cultural prosperity, and sustainable development [2,3]. Museums in cities play a vital role in both universal education and lifelong education. Hooper-Greenhill et al. [4] point out in "Museums and Social Inclusion: The GLLAM Report" that exhibitions and educational outreach activities can promote egalitarianism in social education. In addition, to assist school children to learn in a contextualized, non-formalized, and spontaneous way and provide a learning environment that is different from the school education model, the museums are capable of contributing to social education and culture. Moreover, they can provide professional, public, and diversified educational

resources for socially disadvantaged individuals, groups, and organizations, and enhance the competitiveness of the national society [5].

An exhibition "learning sheet" is an educational material that combines the functions of guiding visitors for reading, exhibition participation, interaction, observation, reflection, debate, and even advertisement, and has been widely implemented in major museum exhibitions around the world. A quality digital learning sheet design can enhance effective information transfer, educational learning, learning motivation and outcomes; while a poor quality digital learning sheet design may eliminate the characteristics and values of museum exhibitions, educational benefits, and could mislead users in the wrong direction of the exhibition learning process and outcomes. Learning sheets are crucial to the promotion of education in exhibitions, but with the pace of modernization, the use of digital learning sheets remains rare, at least in Taiwan where no museums have yet provided "digital learning sheets" for citizens. Although the use of digital learning sheets might have increased some costs, in the digital era, we believe that highly interactive and interesting digital learning sheets can significantly enhance the learning effectiveness of the visitors. We saw the importance of investing in such a study and were motivated to research towards the goal of how museums can design effective digital learning sheets [6].

The National Taiwan Craft Research and Development Institute (hereinafter, NTCRI), established for research, education, exhibition, collection and preservation, is located in the central part of Taiwan, where the craft industry is flourishing and is an important representative of Taiwan's regional museums. In this study, we chose the "Exhibition of Taiwan's Woodcrafts" organized by the NTCRI as a case study. In the first stage of the study, we first conducted participatory observation method to understand the process of the exhibition curation and the establishment of educational purposes. In the second stage, we explored the effects of different strategies for designing digital learning sheets and their impact on visitors' learning motivation and outcomes [7]. In summary, the objectives of this study were to:

1.  Explore how the exhibition objectives correspond and integrate with the educational objectives and educational contents in the exhibition planning framework.
2.  To understand how different digital learning sheet designs affect the learning motivation and learning outcomes of student visitors.

## 2. Literature Review

### 2.1. The Education of Museums

Museums have a history of more than 2000 years, and the earliest museum in the world, The Mouseion of Alexandria, was established in 290 BC by the Egyptian king Ptolemy Soter for royal collections, research, and lecturing purposes [8,9]. With the spread of the knowledge of educational equity, museum scholar Hooper-Greenhill proposed the "post-museum" discourse in 2000, and began to reflect on and criticize the identity of modern museums. According to the post-museum discourse, the nature of the exhibition is believed to be completely open-themed, free, and random, and provide the audience with the knowledge to explore from it [4]. Contemporary museums should be able to respond to the development needs of society, transforming from the object-oriented management mindset of the past to an object-oriented strategy, as well as thinking about the visitor orientation, so that a high level of communication between the audience and the exhibition becomes the main axis of thinking in contemporary museums. The International Council of Museums' (ICOM) definition of a museum [10], adopted by the 22nd General Assembly in Vienna, Austria, on 24 August 2007, is as follows: "A museum is a non-profit, permanent institution in the service of society and its development, open to the public, which acquires, conserves, researches, communicates and exhibits the tangible and intangible heritage of humanity and its environment for the purposes of education, study and enjoyment (2021)". Graeme K Talboys [11] believes that museums play an important role in active cultural interpretation and social communication education. The existence of museums is desired by the general public, and for educational purposes, they carry out operational

behaviors such as acquisition, conservation, maintenance, research, communication, and exhibition. Contemporary museums focus not only on the single aspect of exploration and export of knowledge through acquisition, research, preservation, and demonstration. As early as 1992, the American Association of Museums highlighted in "Excellence and Equality: Education and the Public Dimension of Museums", the importance of social service in terms of museum operations. Since then, the orientation of museums began to focus on their involvement in the development of local communities and on issues of local community events, community identity, and connections as well as the establishment of social networks [12]. Today's museums should think beyond the framework of exhibition in buildings, focusing on their social functions, displaying various records of people and the environment, and reinforcing conceptual aspirations and cultural values, which is why many community museums, regional museums, folk museums, neighborhood museums, school museums, etc., have been established successively. In a message delivered by Alberto Garlandini, President of ICOM on the occasion of International Museum Day in May 2021 [10], he mentioned that the imagination of future museums must be constructed today, and that museum professionals are working for innovative social activities, digitization, new cultural experiences, and complex forms of communication.

Museums are an important asset for local development, and their cultural significance relates to identity, knowledge, emotion, and life in the development of society as well as the intangible cultural heritage of the exhibitions they operate and the dialogue they have with the community. However, the operation of museums is a complex system, which specifically emphasizes cross-disciplinary integration, and postmodern museums may involve various disciplines, such as sociology, culturology, anthropology, semiotics, art education, management, integrated marketing, digital media technology, and other fields; moreover, it combines theoretical exploration and operational practices [13]. The development of museums should be based on the long-term development goals of the local area, city, and even nation.

In the case of museum development in Taiwan, the "Social Education Act" [14] promulgated in 1953 defined museums as one of the social education institutions and could be established by the central or local governments for the purpose of promoting cultural construction, cultivating artistic interests, popularizing technological intelligence, and 18 other social education tasks. In the 2002 "Lifelong Learning Act" [15], museums are classified as one of the types of lifelong education organizations, contributing to lifelong education and social education. In 2016, Taiwan promulgated the "Museum Act" [16] which defined museums in Taiwan as non-profit institutions engaged in the acquisition, conservation, restoration, maintenance, and study of tangible and intangible evidence of human activities and the natural environment, and that are open on a regular basis for the public for the purpose of exhibition, educational promotion, or other uses. In particular, it emphasizes that museums should enhance educational and scholarly functions and improve communication with the public to achieve the purpose of cultural heritage transmission, art promotion, and lifelong learning; and it recommended methods such as "undertaking research related to the museum's purpose or established theme", "transforming research results into content for exhibitions or archived collections", "carrying out education promotion activities or publication of relevant materials" to achieve educational goals.

The emphasis on education is an important transformation in the operation of museums after the 1980s. Museums have become units of public learning as part of a joint effort promoting education [17,18]. Unlike the institutionalized teaching model of schools, museums maintain flexibility and are able to plan different exhibitions and activities for different themes, ethnic groups, and educational purposes, and set specific periods for exhibitions and activities. Museums, in particular, provide visitors with a high degree of experiential participation, making visitors the subjects of exploration, experience, and interaction, actively acquiring diverse and meaningful intellectual cognition and experience from the exhibition, and even generating self-identifying emotional responses, which sometimes have more innovative possibilities with informal education models [19].

From the viewpoint of museum education, the interactive experience model proposed by Falk and Dierking [18] in "The Museum Experience" is one of the important learning processes, which focuses on the interaction between three contexts, which are personal context, social context, and the physical context of the museum, of which interactive experience is formed. As for the participatory experiential learning model, it has been interpreted by Kolb [20], who argues that learning from reflection on doing emphasizes the importance of experiential learning and that the learning process is built on past experiences of life and new experiences. Based on the abovementioned scholars' perspectives, if we were to explain the process of museum knowledge transmission from the viewpoint of cognitive theories, we might benefit from the viewpoint of Constructivist Learning Theory. In other words, the exhibitions provided by museums are characterized in various ways such as materiality, narrativity, sociality, activity, and multimodality. These characteristics can induce visitors to participate actively and meaningfully and to enjoy the context of the exhibition as well as the knowledge it provides [21,22]. Hence, this study argues that the process of knowledge construction by visitors in museums must involve a combination on three aspects: past knowledge and experience, the context of the socio-cultural environment, and new knowledge experienced through participation in exhibitions [23].

### 2.2. Educational Tools for Museums: Learning Sheets

In the process of visiting an exhibition, visitors participate with their thematic interests and concepts according to the attractiveness of exhibits, level of concern, time scheduling, and spatial configuration [18]; however, if we look at it from the perspective of education or information transmission, there are many "noises" in the exhibition environment that might decrease visitors' commitment to the exhibition. The learning sheet in the exhibition serves as an important medium of communication between visitors and the exhibition. Through logical, purposeful, and systematic design, visitors will be able to follow the guidance of the learning sheet and become more engaged in the exhibition. However, learning sheets are supplementary tools for education and learning and the design of learning sheets deserves more attention. If they are designed like flyers, or fail to highlight the context of the exhibition theme or fail to relate to the life experience of the visitors, they will not be of any benefit.

A museum exhibition learning sheet can be designed for different educational purposes, including extended, integrated, exploratory, and activity-based learning sheets. The format can vary according to convenience, operativity, interactivity, and effectiveness, such as booklets, leaflets, folders, and other printed formats, or presented on digital carriers, or even downloaded by visitors from their cell phones. In terms of learning sheet content and question design, Grinder and McCoy [24] suggested four types of question design, which are memory questions, integrated questions, open-ended, and creative questions as well as critical and evaluative questions. Museum curator Hooper-Greenhill [25] emphasizes that visitors should use various senses to learn from exhibitions, and believes that the museum learning model can be divided into five levels, which are: stage 1, sensory and exploration, stage 2, discussion and analysis, stage 3, memory and comparison, stage 4, deep thinking from different cultural backgrounds, and stage 5, cross-field interaction and application. If we examine the four stages of cognitive development, which are proposed by Piaget, the senior of elementary school to the middle school level can already adopt an egocentric viewpoint and they are capable of hypothesizing, interpreting, reasoning, and systematically organizing, comparing, and solving knowledge learning problems. Hooper-Greenhill [25] also proposed three aspects of the question design of the learning sheets, which are 1. a sensory exploration of physical objects, 2. memorization, comparison, and integrative association, and 3. problem discussion, analysis, and integrative comparison and commentary. This perspective also offers important guidelines for the design of the digital learning sheets.

### 2.3. Perspectives from Learning Theory

Learning theory provides abundant insights into the practical work of museums in planning exhibitions. This article explores the various possibilities of museum exhibitions for visitor learning from three perspectives of learning theory. The three perspectives include "constructivist learning theory" which explores the learning resources available to visitors at the exhibition. The "educational goal theory" discusses the different components of education and learning goals that the exhibition can provide. The "learning motivation theory" focuses on how the exhibition enhances various learning motivations, attitudes, and behaviors of visitors.

#### 2.3.1. Constructivist Learning Theory

Museum exhibitions provide visitors with an informal, spontaneous, and enjoyable learning environment. Exhibition curators must consider how to present the exhibited objects' content, knowledge, spirit, and affection through the integration of different methods, media, space, and target audiences, so that visitors can have the opportunity to reconstruct new knowledge frameworks and cognition through the process of visiting and experiencing. Throughout the process, visitors must constantly carry out the tasks of cognitive "assimilation" and "adjustment" of cognitive reconstruction through the information received and interpretation strategies of Prediction, Observation, and Explanation (also known as POE strategies) [26,27]. We can thus conclude that museums provide important learning resources for situated learning just as Lave and Wenger [28] further pointed out that situated learning can be a result of the interaction of activity, context and culture. In addition to the process of personal knowledge construction, museum exhibition design can also create the benefits of cooperative learning through collaboration, work sharing, communication, and debating among visitors under the guidance of planned social interaction [29].

Lev S. Vygotsky (1896–1934) and Jean Piaget (1896–1980) are praised as the most important scholars of the constructivist learning theory in the late twentieth century, and their ideas about the cognitive construction of learning have received much attention in museology. Vygotsky's theory of social construction emphasizes the significance of social activities and three important points made by Vygotsky are critical references for museum exhibitions. Firstly, the "Egocentric Speech" or so-called inner-dialogue, as Vygotsky argues, is beneficial in guiding children's thinking and actions in their learning process. Therefore, if the process of visiting an exhibition can induce children to state their inner thoughts, express and discuss them, it may become a strategy conducive to learning. Secondly, the "Zone of Proximal Development (ZPD)", refers to the gap between children's current knowledge level and the knowledge level they can achieve after being taught by others. Lastly, the "Scaffolding", which is the importance of the instructor in the learning process as suggested by Vygotsky, who believes that a teacher or a more capable peer who provides appropriate assistance will help children's learning [23,30,31].

In Piaget's Genetic Epistemology, the internal learning process is described in terms of "Organization" and "Adaptation". The "Organization" refers to how people organize their experiences and knowledge into logical combinations and define the relationship between each other logically so that people can complete their work efficiently when facing difficulties. "Piaget believes that cognitive development is a cumulative process that builds on existing knowledge (or Schemata) and continues to construct larger or deeper knowledge structures. Piaget considers this as a "theory of self-regulation" in which coordination can be divided into two processes: "assimilation" and "accommodation". In addition, Piaget further divided the cognitive development of children and adolescents into four stages, namely: "Sensory-Motor", which is about 0–2 years old, and is characterized by sensory and motor functions, mostly instinctive reflex behaviors; "Preoperational", which is about 2–7 years old, in which stage children can gradually express and use symbols verbally, but their thinking ability is not fully logical yet; "Concrete Operational" is about 7–11 years old, in which stage they can think concretely and have the ability to think in reverse; "Formal Operational" is about 11–15 years old, at which time they can do abstract

thinking. They can solve problems in a hypothesis-tested scientific way, and think through logical rules [32–35]. Some museums have developed based on Piaget's learning theory, such as the Boston Children's Museum, San Francisco Children's Discovery Museum, and the Children's Creativity Museum. Thus, museums must consider not only the theme and content of the exhibition, but also the presentation methods and media, the age of the visitors, the cognitive schemata, the learning background, and even the educational system and system implemented by the society as a whole, to provide appropriate exhibition.

### 2.3.2. Perspectives from Educational Objectives

From an educational point of view, the designing of educational content is often considered from a "goal-oriented" and "process-oriented" perspective. A "goal-oriented" educational content design emphasizes the outputs and outcomes of learning and the goals that the educational content is intended to achieve are predetermined before the educational content is designed. On the contrary, "process-oriented" education emphasizes the process of learning, experience, and inquiry as well as the level of engagement of the learners. The American educator Tyler proposed the principles of content design and the Tyler Evaluation Model in his book *Basic Principles of Curriculum & Instruction*, which reminded one of the importance of establishing educational goals, selecting learning experiences, organizing learning experiences, and conducting the evaluation [36]. Such a model provides an important reference reminder for exhibition planning.

In 2015, the Taiwan National Academy for Educational Research promulgated the "curriculum guidelines of 12-year basic education general guidelines: core competency development handbook" to provide a reference for education in various fields. "Core competency" as defined in the handbook refers to an ability to meet various needs in life, including the ability to use knowledge, cognition, and skills, as well as attitudes, affections, values, and motivations. In the manual, it is emphasized that learning performance in achieving educational goals should be examined and evaluated along three dimensions: the "cognitive process dimension", the "affective dimension", and the "psychomotor dimension". These three learning performance dimensions are based on Bloom, Krathwohl, and Simpson's theoretical division of teaching objectives into three domains: cognitive, affective, and psychomotor domains [37–39].

A research team led by Benjamin Samuel Bloom (1913–1999) published *Taxonomy of Educational Objectives: The Classification of Educational Goals. Handbook I: Cognitive Domain* in 1956. "Cognitive Domain" provided an important reference for educators from the perspective of learning objectives and assessment, and its classification system divided the cognitive domain into six levels according to different levels of performance. Forty-five years later, Anderson and Krathwohlh published "A Taxonomy for Learning, Teaching, and Assessing: A Revision of Bloom's Educational Objectives", in which the cognitive objectives were divided into the Knowledge Dimension and the Cognitive Process Dimension. The Knowledge Dimension is divided into four categories: Factual Knowledge, Conceptual Knowledge, Procedural Knowledge, and Meta-cognitive Knowledge, reminding educators and curators of the importance to think about what to teach when planning the content of education; whereas the Cognitive Process Dimension is divided into six categories: Remember, Understand, Apply, Analyze, Evaluate, and Create, reminding educators of the significance of how educational content could motivate learners of retention and transfer knowledge [39].

### 2.3.3. Learning Motivation and Outcomes

Museums feature the qualities of informal education, including the following characteristics: (1) visitors are free to choose their learning ways and means in an informal education context; (2) it is a place without the pressure of assessment and competition as in school education; (3) the learners' appreciation of learning can be supported by the exhibition environment and information. In museum exhibitions, visitors have the power

of free-choice learning, choosing and controlling the content and methods of learning based on their interests and needs [18,40].

Museums play a role as educators in society. Hooper-Greenhill [41] suggested that modern museums are crucial in an inclusive society, supporting the lifelong learning of different groups. Therefore, the activities and exhibitions planned by museums must interact with the public in an adequate and age-appropriate educational manner. The services that museums can provide through different information and computer technologies should be improved; to provide an environment that enables audiences to learn, form motivation and generate interest in learning [4,42,43]. Scott [44] regards museum resources, displays, and related educational activities as "products" that museums provide to the public, and emphasizes the idea that learning outcomes should be valued as an educator. In museums, learning outcomes refer to the various tangible and intangible takeaways that visitors receive through the "products" of the museum. Therefore, to examine the degree to which visitors acquire from the various types of "products" offered by the museum, the assessment of learning motivation and learning outcomes is used as evidence to observe the progress of these visitors [45,46].

The motivation to learn is the driving force that causes learners to engage in learning behaviors. Museum exhibitions that make good use of learning sheets as a tool to guide visitors through the learning process and focus may inspire visitors to explore the contents of the exhibition. In general, learning motivation can be divided into Intrinsic Motivation (or regarded as personal variables), which is the drive to learn and it is promoted by the learner's intrinsic needs, such as the desire to learn; and Extrinsic Motivation (or regarded as environment variables), which is the desire to learn due to external environmental stimuli, inducements, and guidance [47]. Keller [48] once used the ARCS motivation model to explain people's motivation to learn. From the viewpoint of museum exhibition education, the "A" stands for attention, which means to consider the attractiveness of the theme, content, aspiration, and form of the exhibition whether it catches visitors' attention and curiosity. The "R" stands for "relevance" which means to consider how the content of the exhibition may engage visitors' personal needs, life, and produce meaningful relevance. "C" stands for confidence that the ability of visitors should enable them to participate in various learning activities in the exhibition with moderate difficulty and challenge. "S" stands for "satisfaction", which is the satisfaction that the exhibition provides, in terms of the pleasant experience, spiritual enhancement, and knowledge learning satisfaction that visitors can get from the process of visiting the exhibition [49,50]. In this regard, Dörnyei [51] points out that the satisfaction of learning is fundamental to the motivation of learning because it confirms that the learner's efforts and the whole learning process are directed towards a goal, purpose, and value.

## 3. Study Methodology

This study was conducted in two stages for the purpose of examining the impact of different digital learning sheet design strategies on the learning outcomes of visitors to the exhibition. In the first stage of the study, we participated in the process of planning, designing, and setting up a professional exhibition by using the participatory observation method to analyze how the people in practice operate and sort out the curatorial model. The second stage of the research is based on the research hypothesis proposed in this study. The methodology of the two stages of the study is explained as follows.

### 3.1. Framework and Hypotheses

This study examines whether the design strategies of different digital learning sheets have an impact on the learning outcomes of the school children who visit the exhibition. Drawing from the literature review, this study chose the principles of learning sheet design suggested by Hooper-Greenhill to design the digital learning sheets for the exhibition and to provide the children who visited the exhibition with digital learning sheets on their iPads. Such a learning sheet is different from the conventional paper-based format and

will provide more interactive, audio-visual, and hyper-linked information to give school children more diversified, interesting, and intimate ways to learn. For the purpose of this study, the following research framework diagram (Figure 1) is proposed based on the previous discussions of related theories.

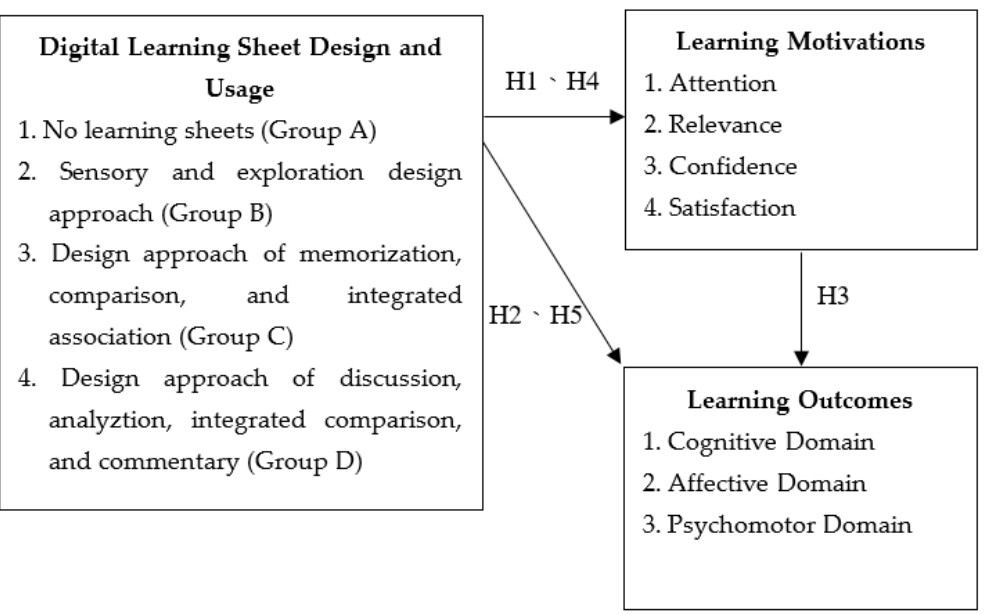

**Figure 1.** Research Framework.

In Figure 1, the relationship between the concepts of "design and usage of digital learning sheets", "learning motivation", and "learning outcomes" is presented in the framework of the study. In the first phase, the researcher participated in the "Exhibition of Taiwan's Woodcrafts" organized by the NTCRI, which was established in 1954 as a regional museum for research, education, exhibition, and collection. This study was based on the educational objectives of the "Exhibition of Taiwan's Woodcrafts", and the study was conducted through expert meetings and the design of digital learning sheets for the exhibition was based on the principles of learning sheet design suggested by Hooper-Greenhill. Students were provided with digital learning sheets on iPads to use in the exhibition. In the second phase, students aged 11 to 15 were invited to the exhibition, and a total of four groups of students participated (groups A, B, C, D). One of the four groups of students are designated as the "control group" (group A) who did not use the digital learning sheets. The other three groups of students were designated as the "experimental group" (groups B, C, and D) who used digital learning sheets designed with different design principles (Appendices A–D). The purpose is to verify the influences of different digital learning sheet design principles on the motivation and learning effectiveness of the students. Details of the study design will be described in the next section.

To investigate the relationship between the concepts of "design and usage of digital learning sheets", "learning motivation", and "learning outcomes", this study was empirically conducted and proposed the following research hypotheses:

**H1:** *The "learning motivation" of the group with digital learning sheets is significantly higher than that of the group without digital learning sheets.*

**H2:** *The "learning outcomes" of the groups with digital learning sheets were significantly higher than those of the groups without digital learning sheets.*

**H3:** *Regardless of the use of digital learning sheets or not, "learning motivation" has a significant and positive impact on "learning outcomes".*

**H4:** *There is a significant difference in the effect of different "digital learning sheet design strategies" on "learning motivation".*

**H5:** *There is a significant difference in the impact of different "digital learning sheet design strategies" on "learning outcomes".*

In the final part of this study, we will comment on the potential benefits and changes in the practice of these digital learning sheet design strategies for museum exhibitions and educational activities, as well as giving suggestions for future use.

*3.2. Research Process*

According to the purpose of this study, two-stage research processes were conducted. The first stage is the researcher's participation in the exhibition curation of the "Exhibition of Taiwan's Woodcrafts", to gain an in-depth understanding of the starting point, motivation, purpose, creative ideas, and exhibition design planning; with in-depth participation of curating the exhibition such as visual design, interactive design, spatial planning, and exhibition service flow, etc., we have summarized the "Art Craft Value Level and Exhibition Structure Relationship Diagram" of the "Exhibition of Taiwan's Woodcrafts" project, and this provides an important basis for the second stage of research. (The exhibition period of "Exhibition of Taiwan's Woodcrafts" is: 30 April 2021 to 17 October 2021.)

In the second stage of the research, the "Exhibition of Taiwan's Woodcrafts" was the subject, and three types of digital learning sheets were designed according to the design principles suggested by Hooper-Greenhill. The main goal of such a design is to develop a digital learning sheet in HTML and JQUERY that can be used on iPads. The digital learning sheets designed by the Institute are different from the printed learning sheets by offering users of digital learning sheets a more user-friendly interactive interface, such as the use of friendly fill-in answers and the use of hyperlinked information (video, text) for extended reading and other interactive content.

The study was conducted by inviting upper elementary school children and junior high school children in central Taiwan, aged between 11 and 15 years old, to be tested from 20 May to 17 September 2021. The subjects were divided into four groups and the testing period did not overlap to avoid interferences. Among the four groups, one of them was the control group of "no learning sheet" (Group A) with 81 participants, while the other three groups participated in different experimental groups with 84 participants in the "sensory and exploration design approach" (Group B), 77 participants in the "design approach of memorization, comparison, and integrated association" (Group C), and 75 participants in the "design approach of discussion, analyzation, integrated comparison, and commentary" (Group D). The number of participants in this study was 317 in total. Before visiting the exhibition, each group of students completed a "pre-test questionnaire" to find out what knowledge they knew about the "Exhibition of Taiwan's Woodcrafts". After the pretest, the students were guided by professional guides on how to operate the digital learning sheets and were guided through the exhibition tour, as well as the reading, filling, and discussion of the digital learning sheets. After each group visit, a "post-test questionnaire" was then conducted to examine the progress of the students' learning outcomes.

*3.3. Research Tools and Analysis Strategies*

3.3.1. Review of Learning Outcomes

To examine the learning outcomes of the four groups of students, A, B, C, and D, after visiting the exhibition and using the digital learning sheet, the study adopted the ARCS model of learning motivation theory and the development of the three dimensions of learning effectiveness: cognitive, affective, and psychomotor. The "pre-test questionnaire" and "post-test questionnaire" were designed with different questions and were given to groups A, B, C, and D to fill out before and after the visit with 4 questions each on the 4 aspects of attention, relevance, confidence, and satisfaction, using Likert's 5-point scale; and 5 questions each on the three aspects of cognition, affect, and psychomotor, using

multiple-choice questions for a total of 31 questions. Among them, we designed 3 reverse questions in order to examine the reliability of the respondents' answers.

The analysis of learning effectiveness in this study was based on the "increase in the number of correct questions" in the cognitive, affective, and skill dimensions of the post-test questionnaire compared to the pre-test questionnaire.

### 3.3.2. Digital Learning Sheet Design

Based on the three different learning sheet design principles proposed by Hooper-Greenhill, this study was conducted to design and produce digital learning sheets on the educational and knowledge contents of the "Exhibition of Taiwan's Woodcrafts". The information contained in the three different learning sheets, such as extended readings and videos, were all consistent in order to avoid research bias caused by information differences. The design process of the digital learning materials was discussed and designed in the form of expert meetings with the participation of curators, university teachers of educational fields, elementary school teachers, junior high school teachers, visual and interface designers, engineers, and our researchers for a total of seven people. The focus and content of each of the three different study sheet designs are briefly described as follows.

"Sensory and Exploration Design Approach" (Group B): The 18 questions in this learning sheet are designed to highlight the process of guiding students to respond to the questions, and to explore and learn through the five senses when visiting the exhibition. It is anticipated that the students may construct an integrated concept in the process of exploration. For example, students are encouraged to explore the weight, aroma, and sound of each type of domestically-produced wood as well as try out wooden furniture, toys, and utensils, while considering the characteristics of the wood and processing techniques behind these exhibits.

"Design Approach of Memorization, Comparison, and Integrated Association" (Group C): This 16-question learning sheet is designed to emphasize the comparison between the new knowledge learned in the exhibition and students' past experiences, and to stimulate expanded thinking and discussion among students and their peers. The discussion of the comparison, correlation, and the cause and effect of different knowledge is also designed in the contents of the learning sheet. For example, students were asked if they could recall any household items made with the techniques of mortise and tenon joints; or questions about their memories of observing natural plants at different altitudes in the past.

"Design Approach of Discussion, Analyzation, Integrated Comparison, and Commentary" (Group D): This learning sheet consists of 16 questions. The design of the contents emphasizes guiding students to observe, analyze, and examine the questions so that they can read further the various information provided in the exhibition and further guide them to have group discussions, comments, and critiques through learning sheets. For example, how can we fulfil our responsibility to protect the environment in our lives? How can different wood tools be used? How does mechanical production compare with manual production? As well as designing a table by yourself and other contents.

### 3.3.3. Reliability Analysis

The total number of questionnaires collected in this study was 317, excluding the invalid questionnaires due to omission or incorrect completion of the reverse question, the total number of valid questionnaires was 312, and the recovery rate was 98.4%. To examine the consistency of the questionnaire content in the "Learning Motivation ARCS" section, the study first examined the internal consistency of the questionnaire using Cronbach's alpha reliability analysis. The results of the reliability analysis showed that the overall reliability of the questionnaire reached 0.916, and the reliability values of each measure were greater than 0.7 (Hee, 2014) (Table 1), indicating that the results of the questionnaire analysis are reliable. Subsequently, Pearson correlation coefficient analysis was chosen for the validation of the study hypothesis (H3) and one way ANOVA was chosen for the mean difference (H1, H2, H4, H5).

**Table 1.** Reliability Analysis of Learning Motivation ARCS Measurements in Four Groups.

| Learning Motivation | Cronbach's Alpha | |
|:---:|:---:|:---:|
| A | 0.857 | |
| R | 0.805 | |
| C | 0.887 | 0.916 |
| S | 0.840 | |

## 4. Research Analysis

*4.1. The First Stage of Study: The Interpretation of the Development of Taiwan Woodcraft Industry with the Concept of Survival, Living, and Philosophy of Life*

In the first phase of this study, the researcher joined the curatorial team of the "Exhibition of Taiwan's Woodcrafts" as consultant and exhibition curator since March 2021, and participated in exhibition meetings, planning, exhibition execution, opening activities, and volunteer guide training. Through participatory observation, we attempt to acquire in-depth understanding of the curatorial team's planning process and the implemented exhibition themes, motives, objectives, as well as creative ideas through exhibition design planning participation such as visual design, interactive design, spatial planning, exhibition service flow, etc. The period of participation was March to May 2021.

During the period of participation, we collected meeting reports, audio recordings, textual materials, and exhibition design plans, and conducted textual analysis to analyze how the professional curatorial team could use the exhibition's contents and presentation methods to communicate wood craft knowledge to visitors.

The study found that the objective of the "Exhibition of Taiwan's Woodcraft" was initially to invite works from the academic, industrial, and artistic fields; later, through the planning process, the curatorial group gradually made clear the knowledge contents and information that the exhibition intended to convey; and then to confirm the exhibited works, exhibition formats, graphics, videos, and interactive details, while gradually organizing the exhibition structure and spatial layout. The curatorial framework of the "Taiwan Woodworking Exhibition" was built upon the Three Extreme Systems of the "Book of Changes" and "Tao Te Ching", written by Lao Tzu, which states that "Tao gives birth to one, one to two, two to three, and three to everything". The central idea was that all things bear the yin and embrace the yang, as they achieve harmony by combining these forces. The curatorial team developed the exhibition from traditional Chinese ideology and divided the value level of crafts into three levels, which are "Tools for Livelihood", "Ways of Living", and "Philosophy of Life", and developed the exhibition objectives and demands into three exhibition areas, such as "Knowledge Learning and Exploration", "Daily Life Application and Connection", and "Spiritual Practice and Pursuit", and planned the exhibition contents of each section accordingly.

In this study, the curatorial structure of the exhibition is reviewed, and the "Art Craft Value Level and Exhibition Structure Relationship Diagram" is proposed as shown in Figures 2–8 below.

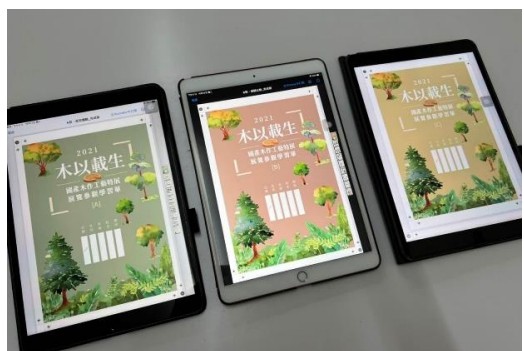

**Figure 2.** In this study, three digital learning sheets were designed based on Hooper-Greenhill's suggestions for the design of learning sheets, and the IPADs were used by visiting students.

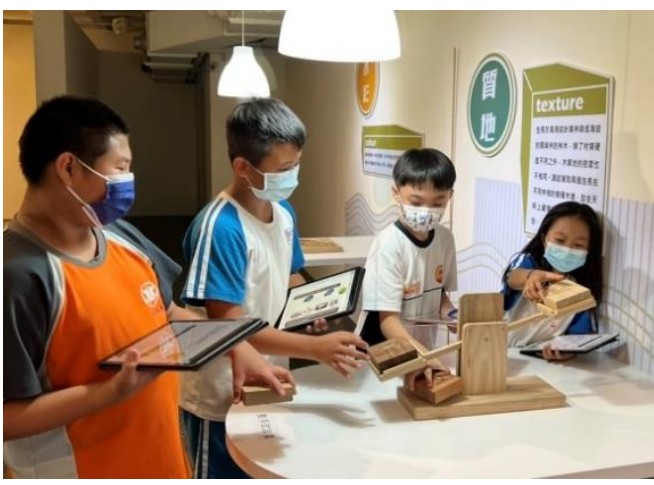

**Figure 3.** The tour guide first guided the students on how to use the digital learning sheets, then guided them step by step through the questions during the tour.

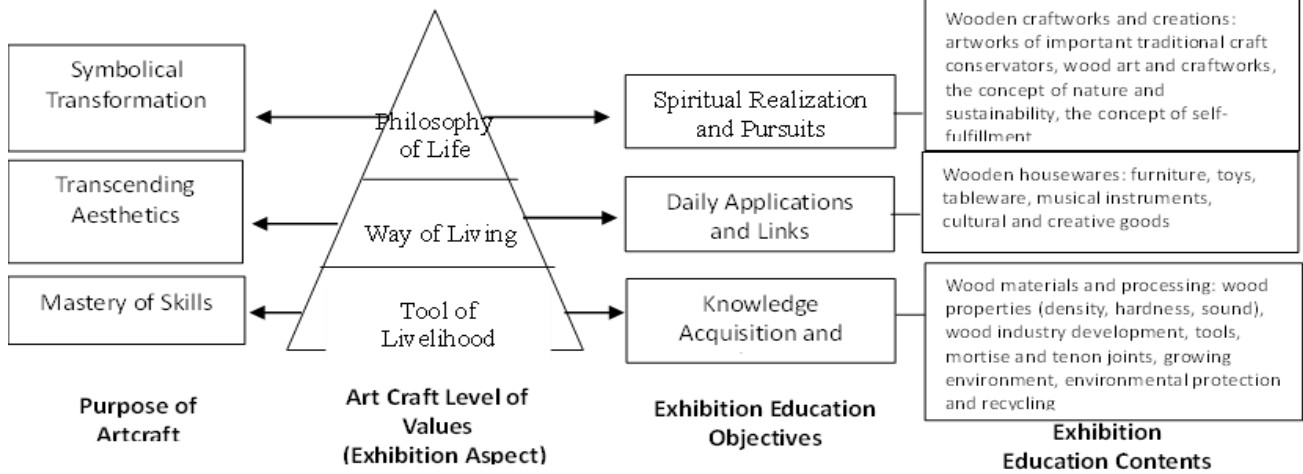

**Figure 4.** Art Craft Value Level and Exhibition Structure Relationship Diagram.

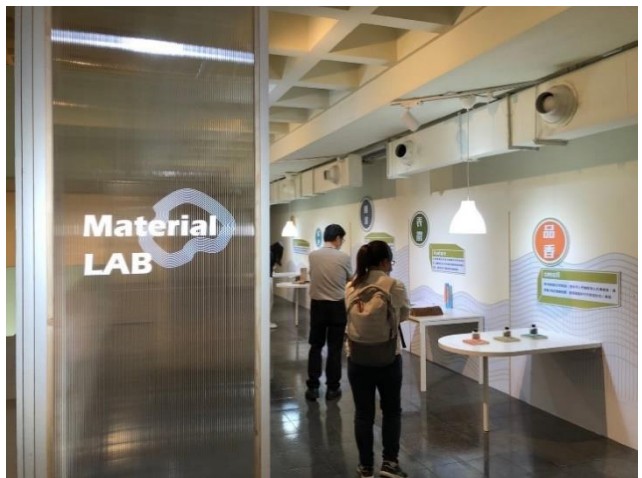

**Figure 5.** Tools of livelihood exhibit section planned with a wood material lab, providing hands-on interaction for the public to learn about various wood properties.

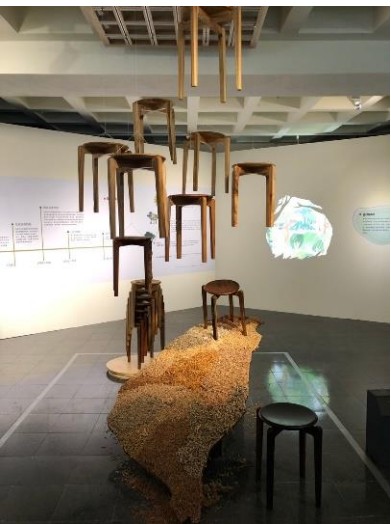

**Figure 6.** Tools of Livelihood exhibition area, including the knowledge of wood industry development.

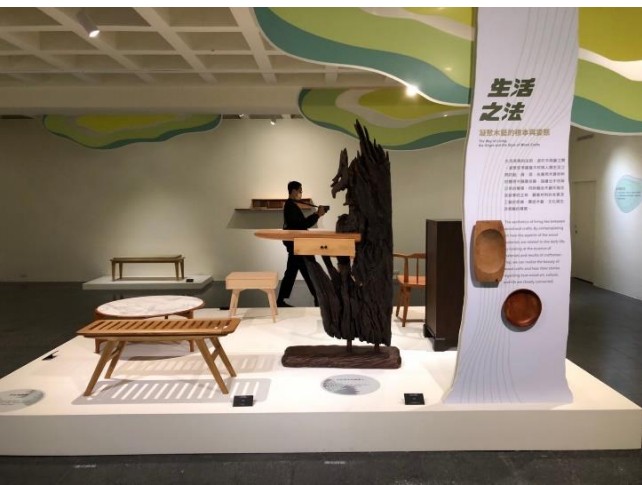

**Figure 7.** The Way of Living section features an introduction to the use of various wood materials in living and innovative designs.

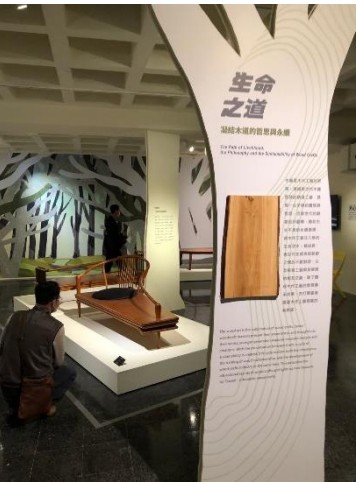

**Figure 8.** Philosophy of Life section, focusing on artists' creations and discourses.

*4.2. The Second Stage of the Study: Verification of Research Hypotheses*

4.2.1. Research Hypothesis 1: The "Learning Motivation" of the Group with Digital Learning Sheets Is Significantly Higher than That of the Group without Digital Learning Sheets

This hypothesis was designed to test whether there was a significant difference in the learning motivation of the students who did not use the digital learning sheets (Group A) compared to the other three groups of students who did use the digital learning sheets (Groups B, C, and D) after visiting the exhibition. In this section, the study analyzed independent samples by *t*-test to see if there were any significant differences between the "learning motivation of group A students" and the "mean of learning motivation of groups B, C, and D students"; the results of the statistical analysis are summarized in Table 2.

**Table 2.** Mean number of "learning motivation" between "no learning sheet" and "using digital learning sheet".

| | Levene's Equality of Variances Test | | The *t*-Test for Equality of Means | | | | | 95% Differences Confidence Interval | |
| --- | --- | --- | --- | --- | --- | --- | --- | --- | --- |
| | F | Significance | t | df | Significance (Two-Tailed) | Average Difference | Standard Error | Lower Limit | Upper Limit |
| Using Equal Variances | 2.043 | 0.155 | −19.263 | 163 | 0.000 | −0.80290 | 0.04168 | −0.72060 | −0.88521 |
| Not using equal Variances | | | −19.186 | 153.010 | 0.000 | −0.80290 | 0.04185 | −0.72023 | −0.88558 |

The results of the analysis showed that group A students had a average learning motivation of 2.7948 after visiting the exhibition, while groups B, C, and D students had an average learning motivation of 3.5977, which was significantly higher than group A students. Table 2 summarizes the results of the independent sample *t*-test, where the F-test was not significant (F = 0.155 > 0.05), so the t-value of "equal variance" = −19.263, where a negative number indicates that the mean of group A is lower than that of groups B, C and D. In terms of significance, it shows that a significant level of 0.000 was reached between the two groups. In other words, this indicates that there is a noticeable difference between the two. This means that hypothesis 1 of this study that the "learning motivation" of the group with digital learning sheets is significantly higher than that of the group without digital learning sheets holds true. This indicates that if learning sheets are used properly in the exhibition, it will help improve the overall learning motivation of visitors.

4.2.2. Research Hypothesis 2: The "Learning Outcomes" of the Groups with Digital Learning Sheet Are Significantly Higher than Those of the Groups without Digital Learning Sheet

This hypothesis examines whether the overall learning outcomes of the students who used the digital learning sheets (groups B, C, and D) were significantly higher than those who did not use the digital learning sheets (group A).

To calculate the learning outcomes of this study, the number of correct questions in each section of the "post-test questionnaire" after the visit to the exhibition was compared to the "front questionnaire" before the visit to the exhibition.

In this section, the study also used independent sample t-testing to examine whether there was a significant difference between the "learning outcomes of group A students" and the "mean learning outcomes of groups B, C, and D students", and the results of the statistical analysis are summarized in Table 3.

**Table 3.** Average of "Learning outcomes" Between "No Learning Sheets" and "Using Digital Learning Sheets".

| | Levene's Equality of Variances Test | | The *t*-Test for Equality of Means | | | | | 95% Differences Confidence Interval | |
| | F | Significance | t | df | Significance (Two-Tailed) | Average Difference | Standard Error | Lower Limit | Upper Limit |
|---|---|---|---|---|---|---|---|---|---|
| Using Equal Variances | 2.428 | 0.210 | −13.219 | 163 | 0.000 | −0.89419 | 0.06765 | −1.02776 | −0.76062 |
| Not using equal Variances | | | −13.161 | 151.415 | 0.000 | −0.89419 | 0.06794 | −1.02843 | −0.75995 |

The statistical results showed that the average number of correct questions improved by 3.0327 for Group A students who did not use the digital learning sheets after visiting the exhibition, while the average number of correct questions improved by 3.9269 for Groups B, C, and D students who used the digital learning sheets, which was noticeably higher.

In Table 3, the T test results of the independent samples were presented, in which the F test was significant (F = 0.210 > 0.05), so that the t-value of "equal variance used" = −13.219, the negative value of which indicates that the average of group A is lower than that of groups B, C, and D. The significance of the two groups reached a significant level of 0.000. In other words, the overall effect of using the digital learning sheets during the exhibition visit significantly improved the learning outcomes of the students, compared to the learning outcomes of those who did not use the digital learning sheets.

4.2.3. Research Hypothesis 3: "Learning Motivation" Has a Significant and Positive Effect on "Learning Outcomes" with or without the Use of Digital Learning Sheets

The third hypothesis was to examine the relationship between "learning motivation" and "learning outcomes", whether there is a positive relationship between learning motivation and learning outcomes for the overall visitation behavior with or without the use of learning sheets.

In this section, the Pearson correlation coefficient was used to compare the mean relationship between the two factors. The results showed a significant positive correlation between "learning motivation" and "learning outcomes", with a significance of 0.000 and a correlation coefficient of 0.404, indicating a moderate correlation between the two factors. It also confirms research hypothesis 3: "Regardless of the use of digital learning sheets or not, 'learning motivation' has a significant and positive impact on 'learning outcomes'. Besides the above-mentioned relationship between the two, the statistical results indicated that the "Exhibition of Taiwan's Woodcrafts" could provide a certain degree of education through the content of exhibition planning, exhibition information, display effect, exhibits of items and service flow, and promote the learning motivation and effectiveness of the visitors.

4.2.4. Research Hypothesis 4: There Is a Significant Difference in the Effect of Different "Digital Learning Sheet Design Strategies" on "Learning Motivation"

In Hypothesis 4, the study examined whether the digital learning sheets designed with different design strategies had varied effects on the "learning motivation" of the students who visited the exhibition. This will help future exhibition curators, learners and designers, as well as formal and informal educators in their design and educational activities, as a reference for their practical experience. This study was conducted by one-way ANOVA with post hoc comparisons using the Scheffe method.

It was found that the three different types of digital learning sheets would have a significant effect on the students' attention, with a significance of 0.000 (Table 4). In this study, a post-comparison using the Scheffe method showed that there was a significant

difference between the mean number of Attentions obtained from the three different learning sheets, with Group B having the highest mean among the three groups (Table 4), which means that the digital learning sheets designed with the "sensory and exploration design approach" would promote the highest Attentions among the children.

**Table 4.** Post hoc comparisons of the ARCS variance analysis for the four groups of B, C, and D students (Scheffe).

| Post Hoc Comparisons of the Attention Variance Analysis of Students in Groups B, C, and D | | | | |
|---|---|---|---|---|
| Group | N | Alpha = 0.05 Subset | | |
| | | 1 | 2 | 3 |
| B Group | 84 | 3.8214 | | |
| C Group | 77 | | 3.4221 | |
| D Group | 75 | | | 2.9933 |
| Significance | | 1.000 | 1.000 | 1.000 |
| **Post Hoc Comparisons of the Relevance Variance Analysis for Students in Groups B, C, and D** | | | | |
| Group | N | alpha = 0.05 Subset | | |
| | | 1 | 2 | |
| B Group | 84 | 3.7589 | | |
| C Group | 77 | | 3.4643 | |
| D Group | 75 | | 3.3533 | |
| Significance | | 1.000 | 0.317 | |
| **Post Hoc Comparisons of the Confidence Variance Analysis for Students in Groups B, C, and D** | | | | |
| Group | N | alpha = 0.05 Subset | | |
| | | 1 | 2 | |
| B Group | 84 | 3.7946 | | |
| C Group | 77 | 3.711 | 3.711 | |
| D Group | 75 | | 3.5733 | |
| Significance | | 1.000 | 0.534 | |
| **Post Hoc Comparison of the Analysis of Satisfaction Variance among Students in Groups B, C, and D** | | | | |
| Group | N | alpha = 0.05 Subset | | |
| | | 1 | 2 | |
| B Group | 84 | 3.872 | | |
| C Group | 77 | | 3.6461 | |
| D Group | 75 | | 3.6233 | |
| Significance | | 1.000 | 0.903 | |

Further comparison with the Scheffe method showed that the mean of Group B was significantly higher than that of Groups C and D among the three different learning lists, while there was no significant difference between Groups C and D in statistical analysis. Examining the impact of the three different types of digital learning sheets on students' Relevance, the results showed an equally significant impact, with a significance of 0.000 (Table 4).

Further examining the effects of the three different types of digital learning sheets on students' Confidence, statistical analysis revealed that the three different types of digital learning sheets resulted in a significant difference in students' Confidence, with a

significance of 0.015. After a comparison of the Scheffe method, there was a significant difference between Group B and Group D. The mean Confidence feedback for Group B was 3.79, which was significantly higher than the mean of Group D, which was 3.57 (Table 4).

Finally, the study examined the effects of the three different types of digital learning sheets on the students' Satisfaction, and the statistical result reached a significant level of 0.000. A post hoc comparison in the Scheffe method revealed that the mean Satisfaction feedback of the students who used the "sensory and exploration design approach" digital learning sheet was significantly higher than the other two groups. (Table 4)

In summary, when students in groups B, C, and D used digital learning sheets designed with different learning sheet design strategies while visiting the exhibition, they were motivated to learn at different levels and their impact was highly significant. That is, research hypothesis 4 that "there is a significant difference in the effect of different "digital learning sheet design strategies" on "learning motivation", is valid.

From the analysis of the four aspects of learning motivation, namely, Attention, Relevance, Confidence, and Satisfaction, the digital learning sheets designed according to the "sensory and exploration design approach" were more effective than the other two groups of students. Therefore, it can be inferred that Taiwanese students aged 11 to 15 appeared to be more motivated to participate in the exhibition through sensory experience and exploration, whether by exhibiting objects in an interactive format or by using learning sheets to guide them to use their senses to work on, experience, experiment, and examine the exhibition content.

In contrast, the study also observed that among the three different strategies for designing the digital learning sheets, the lowest motivation feedback was found among Group D students who used the digital learning sheets' "design approach of discussion, analyzation, integrated comparison and commentary". The reason could be explained by the fact that in Taiwan's traditional elementary and junior high school education, it is more common to see teachers giving lectures unilaterally, which is criticized by the education sector in Taiwan as a "fill-in-the-duck teaching method". As a result, students do not have sufficient opportunities to discuss, evaluate, discuss, and synthesize issues during their school learning. In the course of this study, it was observed that at least half of the students were afraid to express their opinions during group activities due to timidity, shyness, or concern for the judgment of others and peers. The students in Group D were unable to focus their minds on the learning of the knowledge domain and were more concerned with social issues, resulting in lower motivation than the other two groups.

### 4.2.5. Research Hypothesis 5: There Is a Significant Difference in the Impact of Different "Digital Learning Sheet Design Strategies" on "Learning Outcomes"

In Hypothesis 5, the study examines whether different digital learning strategies have a significant impact on the learning outcomes of the children who visit the exhibition, and the three dimensions of Cognitive, Affective, and Psychomotor are examined, respectively. The marking system for "learning outcomes" was also based on the number of correct questions in the "post-test questionnaire" compared to the "pre-test questionnaire".

Based on the results of the analysis of variance, it was found that the different design strategies of the digital learning sheets significantly affected the learning effectiveness of the Cognitive component, with a significance of 0.010. From the post hoc Scheffe method comparison, the mean number of correct answers on the Cognitive section was 4.27 for Group B, which was significantly higher than the mean number of correct answers on the Cognitive section of 3.86 for Group C (Table 5).

Next, the results of the analysis of variance showed that there was no significant difference in the mean number of correct answers for the Affective section among the students using different digital learning sheets, with a significance of 0.466. The mean number of correct answers for Groups B, C, and D, respectively, was 3.83 for Group B, 3.99 for Group C, and 3.95 for Group D. The statistical results show the significance that

different strategies for designing the digital learning sheets do not produce significant differences in the Affective of learning outcomes.

**Table 5.** Post hoc comparison of the Cognitive and Psychomotor variance analysis of students in groups B, C, and D (Scheffe).

| Post Hoc Comparison of Cognitive Variance Analysis of Students in Groups B, C, and D | | | |
|---|---|---|---|
| Group | N | Alpha = 0.05 Subset | |
| | | 1 | 2 |
| B Group | 84 | 4.27 | |
| C Group | 75 | 4.12 | 4.12 |
| D Group | 77 | | 3.86 |
| Significance | | 0.538 | 0.165 |
| **Post Hoc Comparisons of Psychomotor Variance Analysis for Students in Groups B, C, and D** | | | |
| Group | N | Alpha = 0.05 Subset | |
| | | 1 | 2 |
| B Group | 84 | 4.11 | |
| C Group | 75 | | 3.57 |
| D Group | 77 | | 3.57 |
| Significance | | 1.000 | 1.000 |

Finally, regarding the Psychomotor component of learning outcomes, the results of the analysis of variance showed that the three different digital learning sheets significantly affected the Psychomotor learning outcomes of the students with a significance of 0.000 (Table 5). The results of this study, which were further compared using the Scheffe method, showed that Group B students who used the "principle of sensory exploration of physical objects" digital learning sheet had significantly higher Psychomotor learning outcomes than the other two groups (Table 5).

Summing up the results of the above study, research hypothesis 5, which is that "there is a significant difference in the impact of 'different digital learning design strategies' on 'learning outcomes'" is partially valid. In addition to the Affective learning outcomes, the cognitive and psychomotor learning outcomes were better for children who used the Design Principles for Sensory Exploration of Objects (Group B) digital learning sheet. We concluded that interactive learning through sensory experiences in exhibitions for students aged 11 to 15 years in Taiwan could create better comprehension and memory outcomes for students. This result echoes the view of Kimche [52], the former executive president of the American Association of Science and Technology Centers, that the most effective way to learn is for people to explore things that interest them in a personal and experiential way. Therefore, the museums could consider providing visitors with effective learning experiences, sensory exploration, group cooperative learning, and interpersonal intelligence development [53]. This study result would be beneficial in promoting learning attitudes and effectiveness, and such results could serve as an important reference for future exhibition planning and learning sheet design.

## 5. Conclusions and Suggestions

As a provider of informal education, museums play a critical role in the lifelong education of society. It is important to know that the operation of regional museums is to consider ways to revitalize local industries, promote the artistic life of residents, assist in the development of education, and even more importantly, provide a place with the functions of tourism, leisure, entertainment, and education.

A quality themed exhibition provides the community with ample opportunities for knowledge learning. This study attempts to provide comprehensive empirical research of the design of digital learning sheets for museum exhibitions. Therefore, the learning sheet design approach, proposed by museum curator Hooper-Greenhill, is adopted as the foundation. Digital technology has been applied to integrate more interactive, extended reading, and fun possibilities into the design of three digital learning sheets.

This study chose the "Exhibition of Taiwan's Woodcrafts", which was organized by the National Taiwan Craft Research and Development Institute (NTCRI), as a case study to examine whether the use of different types of digital learning sheets had a significant impact on the "learning motivation" and "learning outcomes" of the students when they visited the exhibition. The study results are promising. In the study, we found that for Taiwanese students, the benefits of the digital learning sheets designed using the "sensory and exploration design approach" were better than those of the other two groups in terms of motivation and learning outcomes. The study concluded that the digital learning sheets designed with the "sensory and exploration design approach" could guide the students to engage actively in the context of the exhibition. Such a design strategy encourages the use of different senses, such as sight, smell, and touch, to deepen the memory of the exhibition content. This is the main reason why we believe that the digital learning sheets designed with the "sensory and exploration design approach" can lead to the best learning outcomes for the learners. The results will contribute significant references for future curatorial teams as well as for digital learning unit designers and educators in their design and educational implementation. In addition, this study also reminds us that in both formal and informal education contexts, the teaching strategies implemented by the educators must be more in line with the learning styles and patterns of the students in order to achieve better learning outcomes.

Finally, digital learning is a rising future trend. However, the use of digital learning in museums remains rare. We believe that digitization, convenience, and interactivity are the development directions and are a necessity for future lifestyles. This study focuses mainly on the digital learning sheet design strategies of museum exhibitions, and it also adopted quantitative questionnaires to examine the effectiveness of learning. Nevertheless, this study still offers directions for future studies to explore the various possibilities of digital learnings in museums and a possibility for further research in a more in-depth qualitative research approach. We hope that the study result might as well contribute to the digital transformation of museums so that there might be more possibilities of museums in providing diversified services in exhibitions, education, and learning.

**Author Contributions:** Conceptualization, T.-L.C. and C.-S.H.; methodology, Y.-C.L.; software, Y.-C.L.; validation, T.-L.C., Y.-C.L. and C.-S.H.; formal analysis, Y.-C.L.; investigation, Y.-C.L.; resources, T.-L.C.; data cu-ration, T.-L.C. and C.-S.H.; writing—original draft preparation, Y.-C.L.; writing—review and editing, Y.-C.L.; visualization, T.-L.C. and C.-S.H.; supervision, T.-L.C. and C.-S.H.; project administration, Y.-C.L., T.-L.C. and C.-S.H. All authors have read and agreed to the published version of the manuscript.

**Funding:** This research received no external funding.

**Institutional Review Board Statement:** Ethical review and approval were waived for this study due to this study does not involve human experimentation, psychological stress, or anything that might cause physical or psychological damage to the students.

**Informed Consent Statement:** Informed consent was obtained from all subjects involved in the study.

**Data Availability Statement:** Not applicable.

**Conflicts of Interest:** The authors declare no conflict of interest.

## Appendix A. Pre-Test and Post-Test Questionnaires (The Implementation Is in the Chinese Version)

- Pre-test Questionnaires

Class : ______________ Name : _______________ Date : _____________

**Welcome, students!**
We are about to visit "Into The Woods - An Exhibition of Taiwan's Woodcrafts" presented by the National Taiwan Craft Research and Development Institute. Do some of you have some knowledge about Taiwan's wood products, tools, and artworks?
We would like to ask you to answer some questions first, so that we know what you expect from the exhibition.
Please try to answer them in detail.

**Part 1**

|  | Strongly Agree | Agree | Undecided | Disagree | Strongly Disagree |
|---|---|---|---|---|---|
| 1. I'm excited to visit the exhibition later on. | ☐ | ☐ | ☐ | ☐ | ☐ |
| 2. There are plenty of things to check out in the exhibit later that I want to see in detail. | ☐ | ☐ | ☐ | ☐ | ☐ |
| 3. Before coming to see the exhibition, I purposely went to read the books on wood first. | ☐ | ☐ | ☐ | ☐ | ☐ |
| 4. I think the content of the exhibition may be important to me. | ☐ | ☐ | ☐ | ☐ | ☐ |
| 5. I will learn much knowledge that I can apply in my daily life from the exhibition lectures. | ☐ | ☐ | ☐ | ☐ | ☐ |
| 6. I usually do a good job of environmental protection. | ☐ | ☐ | ☐ | ☐ | ☐ |
| 7. I can name at least five kinds of trees | ☐ | ☐ | ☐ | ☐ | ☐ |
| 8. I would like to pay more attention to the issue of wood recycling after watching the exhibition | ☐ | ☐ | ☐ | ☐ | ☐ |
| 9. I know what can be made from wood. | ☐ | ☐ | ☐ | ☐ | ☐ |
| 10. After the exhibition, I think I will read more books about wood. | ☐ | ☐ | ☐ | ☐ | ☐ |
| 11. After seeing the exhibition, I think I would like to know what else is made of wood | ☐ | ☐ | ☐ | ☐ | ☐ |
| 12. After seeing the exhibition, I think I will like the nature subjects more. | ☐ | ☐ | ☐ | ☐ | ☐ |

**Part 2**

1. ( ) When did Taiwan's earliest forestry industry begin to develop? ? (1) Ming Dynasty (2)Qing Dynasty (3) The Japanese Rule Era (4) Early Republic of China Era
2. ( ) Which of the following does not belong to the 5R ecological cycle of national production materials ? (1)Reduce (2)Reuse (3)Redesign (4)Recovery
3. ( ) What is the average altitude of broad-leaved forest? (1) Below 500m (2) 500-2000m (3) 2000-3500m(4)3500 公尺以上
4. ( ) Which of these is recyclable? (1) Wooden furniture (2)Wood Toys (3)Wooden Chair (4) None of the above can be recycled
5. ( ) I know what are made of wood ? (1)Wood Block (2)Guitar (3)Xylophone (4)All of the above answeres.
6. ( ) What are the benefits of man-made forests? ? (1) Reduces pests and diseases (2) Reduces earthquakes (3) Allowing diversity of creatures to live inside (4) Reduces carbon footprint in the production of wood products
7. ( ) If you want to combine the wood, and not using the nail and gluing method, what is this technique called? (1)Bridle Joint (2) Engraving (3) Merging (4) Frame set
8. ( ) Which of the following does the logo in Figure(1) belong to?
   (1) Traceable Agricultural Products
   (2) Certified Agricultural Standards (CAS) certification
   (3) Forest products production traceability barcode (4) Domestic Timber Mark

Figure 1.

9. ( ) If I were to make a percussion instrument out of wood, what characteristics would I look for in wood first? (1) Price (2) Processing (3) Firmness (4) Wood age

●　Post-test Questionnaire

Class : ______________ Name : ______________ Date : ______________

**Hello, students!**
After visiting the exhibition"Into The Woods - An Exhibition of Taiwan's Woodcrafts" presented by the National Taiwan Craft Research and Development Institute," do you have a better understanding of Taiwan's wood, woodworking tools, techniques, as well as household items and artworks?
We would like to ask you to answer some questions to let us know what you have learned about the exhibition and what you have learned from it.
So, please answer them in detail! After answering all the questions, we have a small gift for you!

### Part 1

|  | Strongly Agree | Agree | Undecided | Disagree | Strongly Disagree |
|---|---|---|---|---|---|
| 1. I think the exhibition content is novel and interesting to me | ☐ | ☐ | ☐ | ☐ | ☐ |
| 2. There were many things in the exhibition that I found that would make me want to go and see it in detail. | ☐ | ☐ | ☐ | ☐ | ☐ |
| 3. I have concentrated on seeing what is on display in each place. | ☐ | ☐ | ☐ | ☐ | ☐ |
| 4. After the exhibition, I will want to learn more about it after I go home. | ☐ | ☐ | ☐ | ☐ | ☐ |
| 5. I think the content of the exhibition is what I have learned in school. | ☐ | ☐ | ☐ | ☐ | ☐ |
| 6. I don't think the content of the exhibition has anything to do with my everyday life. | ☐ | ☐ | ☐ | ☐ | ☐ |
| 7. I think most of the things mentioned in the exhibition can be applied in life. | ☐ | ☐ | ☐ | ☐ | ☐ |
| 8. After the exhibition, I felt more concerned about the importance of environmental protection. | ☐ | ☐ | ☐ | ☐ | ☐ |
| 9. After the exhibition, it does not inform me of the characteristics of different woods. | ☐ | ☐ | ☐ | ☐ | ☐ |
| 10. After the exhibition, it makes me pay more attention to the issue of wood recycling | ☐ | ☐ | ☐ | ☐ | ☐ |
| 11. After the exhibition, it reminds me to know which products in my life are made of domestic wood. | ☐ | ☐ | ☐ | ☐ | ☐ |
| 12. After the exhibition, it enables me to know more about the types of trees | ☐ | ☐ | ☐ | ☐ | ☐ |
| 13. After seeing the exhibition, I feel that I have learned something | ☐ | ☐ | ☐ | ☐ | ☐ |
| 14. After the exhibition, I realized that wood can be used to produce various items. | ☐ | ☐ | ☐ | ☐ | ☐ |
| 15. After the exhibition, I imagine what else can be done with wood. Such as : ____________ | ☐ | ☐ | ☐ | ☐ | ☐ |
| 16. Overall, I believe it was quite fruitful to see this exhibition today | ☐ | ☐ | ☐ | ☐ | ☐ |

### Part 2

1. (　) What has happened to Taiwan's timber industry as the United Nations passed the Convention on Biological Diversity? (1) Products cannot be exported, (2) Wooden furniture is becoming increasingly popular, (3) Consumers do not like to use wooden products, (4) Taiwan's timber industry has declined.

2. (　) Which of the following does not belong to the 5R ecological cycle of national production materials ? (1)Reduce (2)Reuse (3)Redesign (4)Recovery

3. (　) Why is it that woods of the same size can have different weights? (1) different density due to growth altitude, (2) jerry-building, (3) some have been exposed to the sun, (4) scale inaccuracy

4. (　) When did Taiwan's earliest forestry industry begin to develop? ? (1) Ming Dynasty (2)Qing Dynasty (3) The Japanese Rule Era (4) Early Republic of China Era

5. (　) Which of these trees grows at an altitude of 2,000 meters? (1) Taiwan White Fir, (2) Japanese Cedar, (3) Taiwan  Acacia, (4) Moso bamboo

6. (　) Which of the following cultural contexts is the most appropriate for the use of wooden furniture and objects? (1) Music appreciation, (2) Tea tasting space, (3) Floral appreciation atmosphere, (4) All of the above

7. (　) With the concept of "green consumption", how would it be beneficial? (1) Environmentally friendly, (2) Durable products, (3) Cheaper prices, (4) Artistic legacye

8. (　) Which of the following themes is not the objective of this exhibition? (1) The tools of livelihood, (2) the sustainability of ecology, (3) the way of living, and (4) the philosophy of life

9. (　) Why do we need to rethink the use of domestic timber? (1) Maintaining sustainable use of resources, (2) Abundant forest resources in Taiwan, (3) Promoting development of forest industry, (4) All of the above

10. (　) What are the characteristics of the President's table known as the "Root Table"?(1) the use of twelve local materials, (2) the use of beech wood to express " achievement of excellence", (3) the carving of Taiwan Lily,　(4) all of the above

11.(　) There were many pieces of furniture that could be joined very securely without the use of nails and glue so what is this technique called? (1) tenon jointing, (2) engraving, (3) merging, (4) framing

12.(　) Which of the following musical instruments can be made of wood? (1) Ukulele, (2) Violin, (3) Xylophone, (4) All of the above

13.(　) The logo in Figure (1) belongs to which of the following?
(1) Traceable Agricultural Products (2) Certified Agricultural Standards (CAS) certification (3) Forest products production traceability barcode (4) Domestic Timber Mark

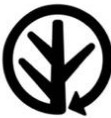

Figure  1.

14.(　) If I were to make a percussion instrument out of wood, what characteristics would I look for in wood first? (1) Price (2) Processing (3) Firmness (4) Wood age

15.(　) Which tool do you think is used to make the leaf petals of the flower so thin in Figure (2)? (1) carving knife, (2) wood saw, (3) planer, (4) chisel

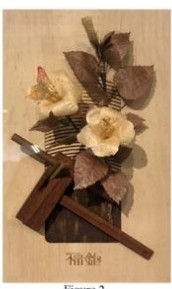

Figure 2.

## Appendix B. Sensory and Exploration Design Approach Learning Sheet (B Group)

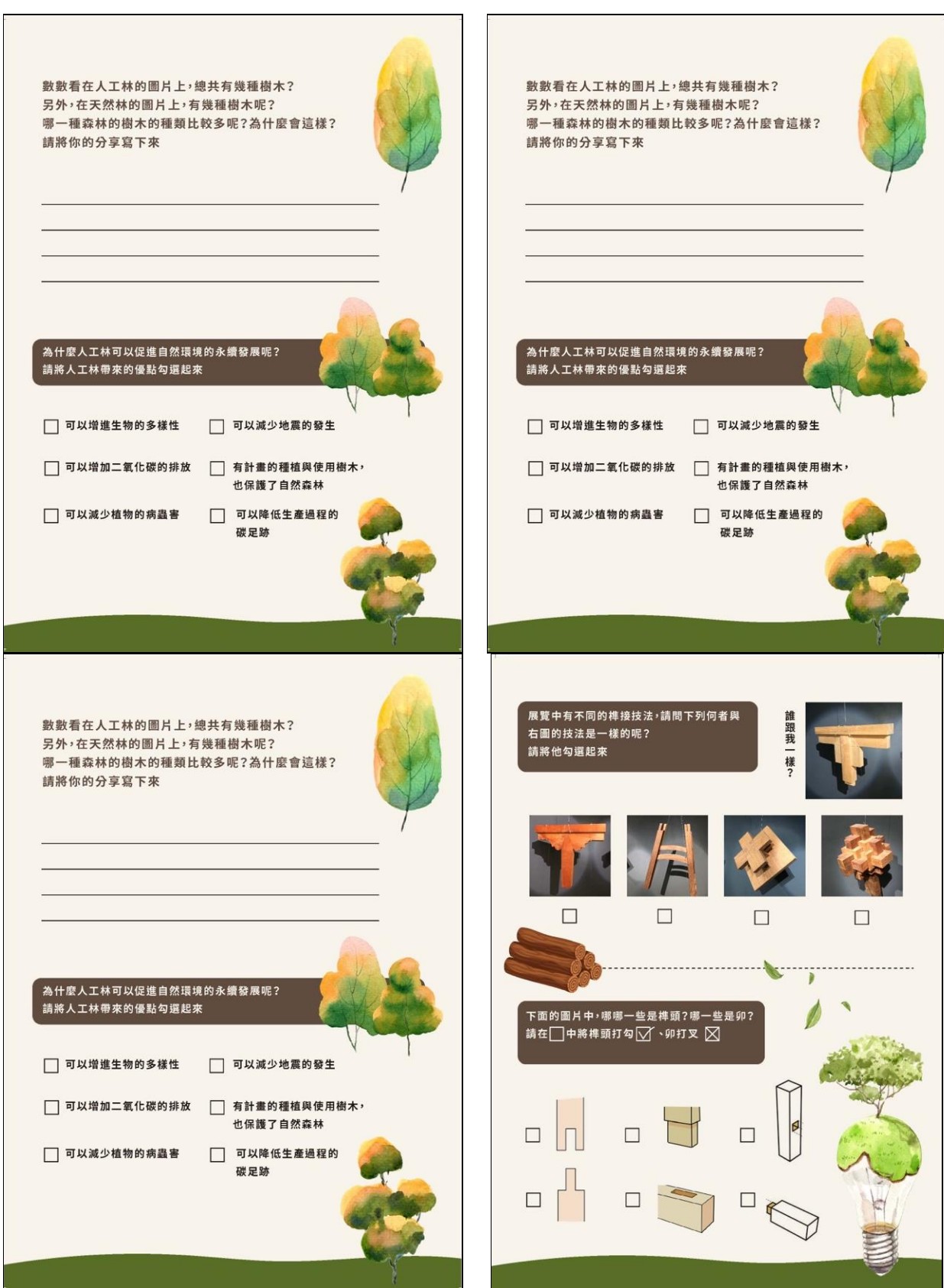

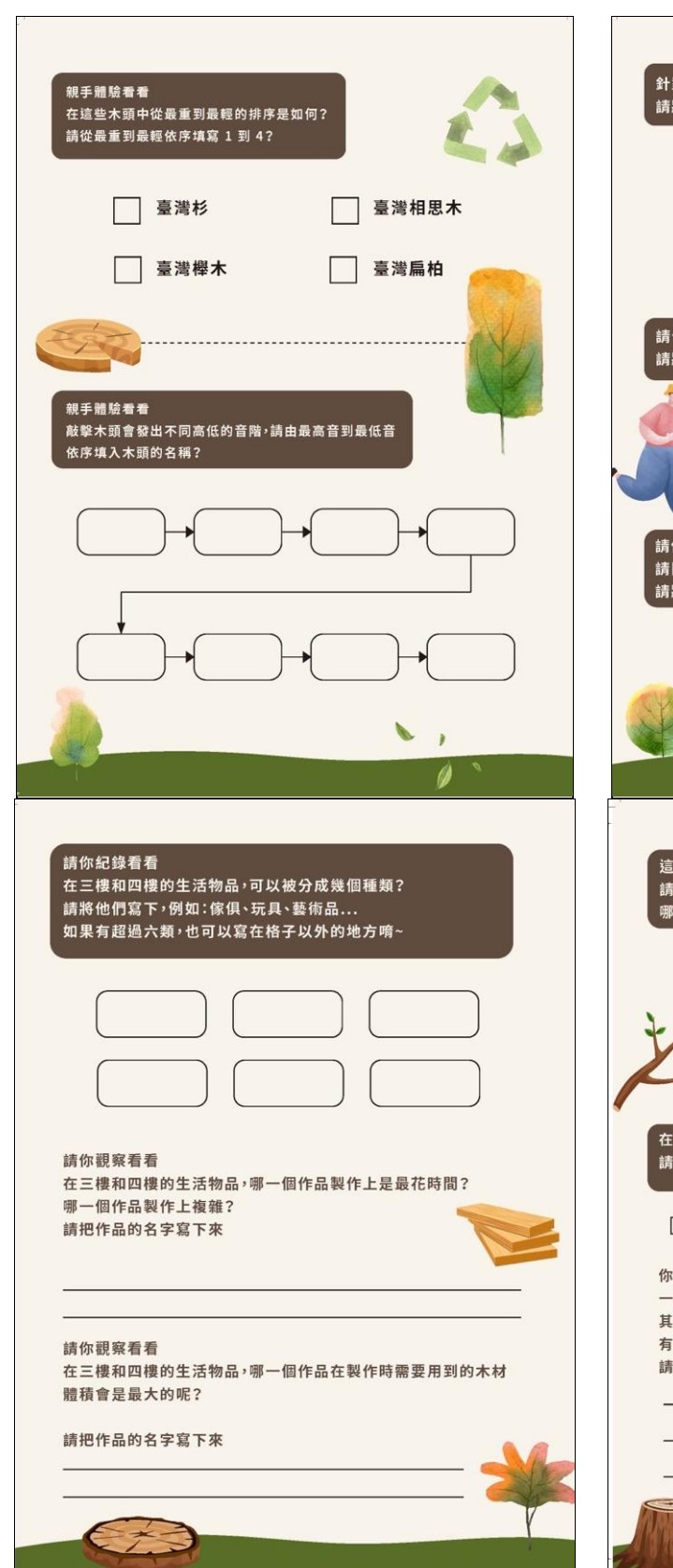

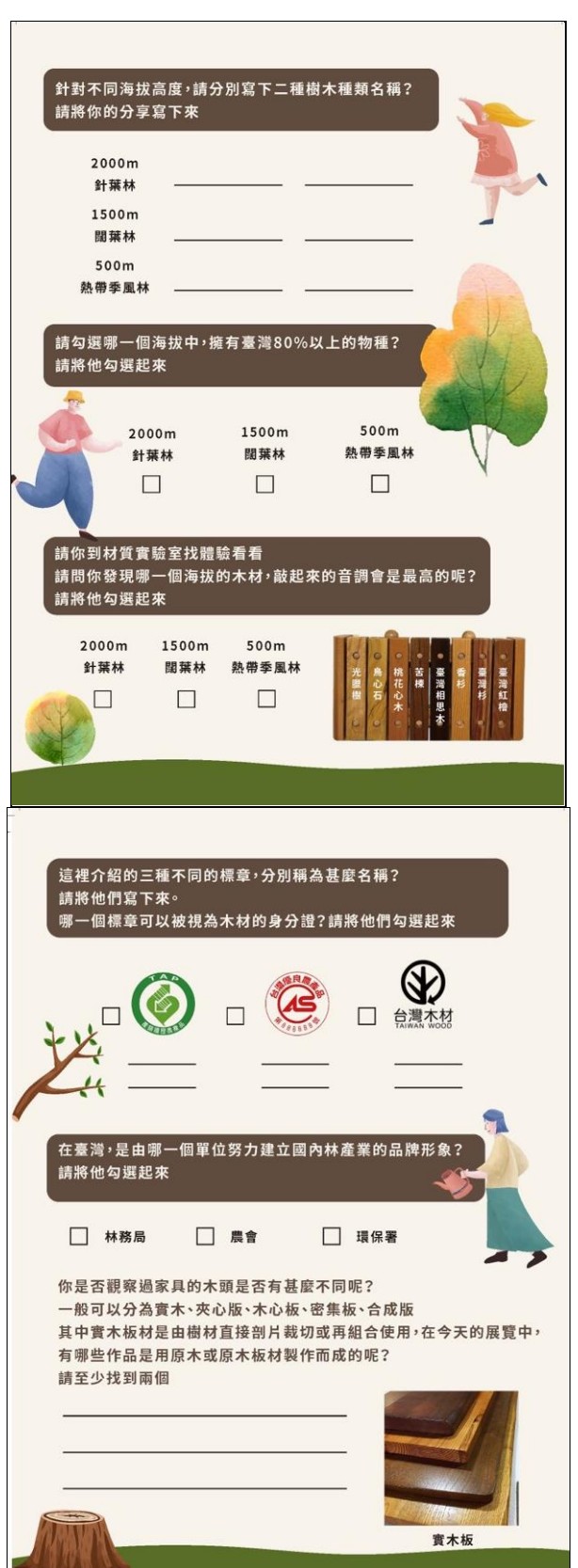

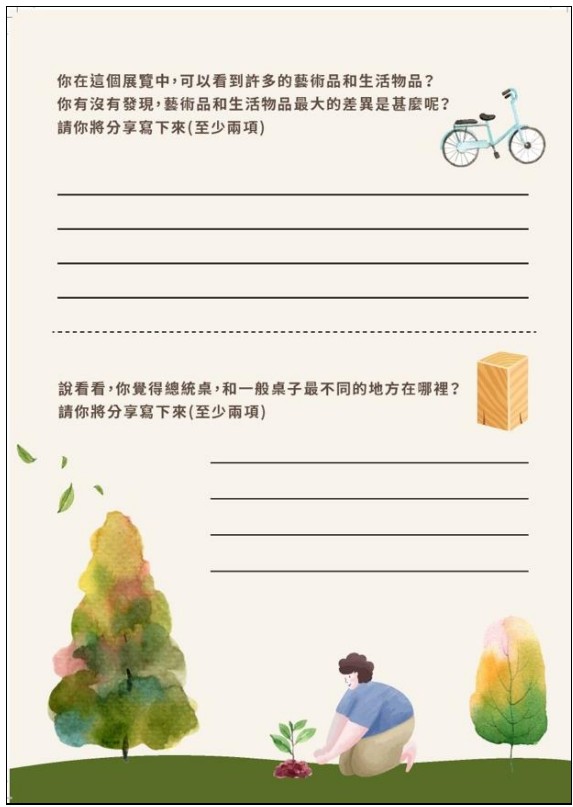

你在這個展覽中,可以看到許多的藝術品和生活物品?
你有沒有發現,藝術品和生活物品最大的差異是甚麼呢?
請你將分享寫下來(至少兩項)

說看看,你覺得總統桌,和一般桌子最不同的地方在哪裡?
請你將分享寫下來(至少兩項)

**Appendix C. "Design Approach of Memorization, Comparison, and Integrated Association" (Group C)**

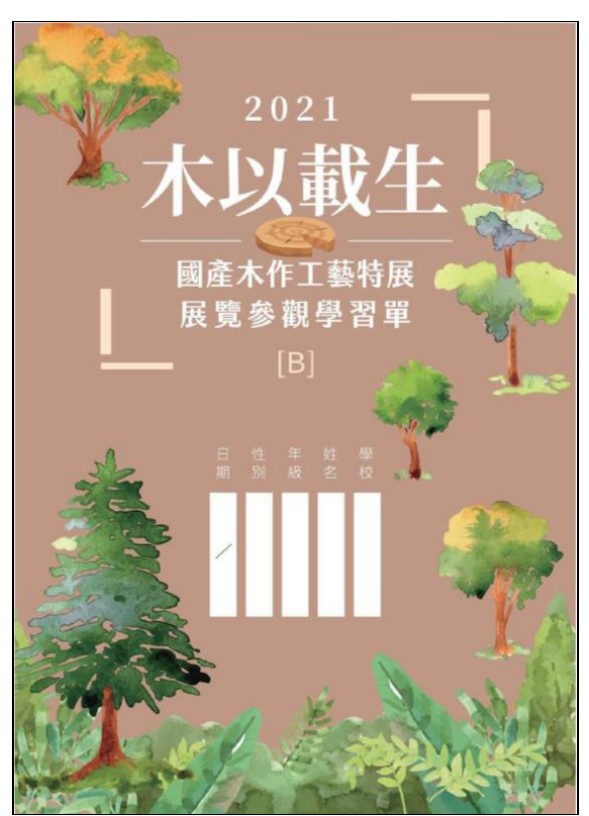

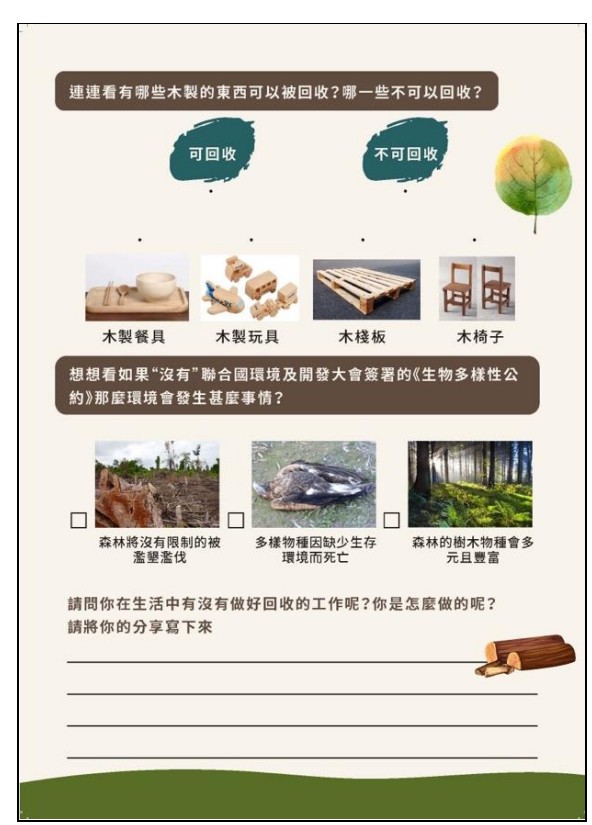

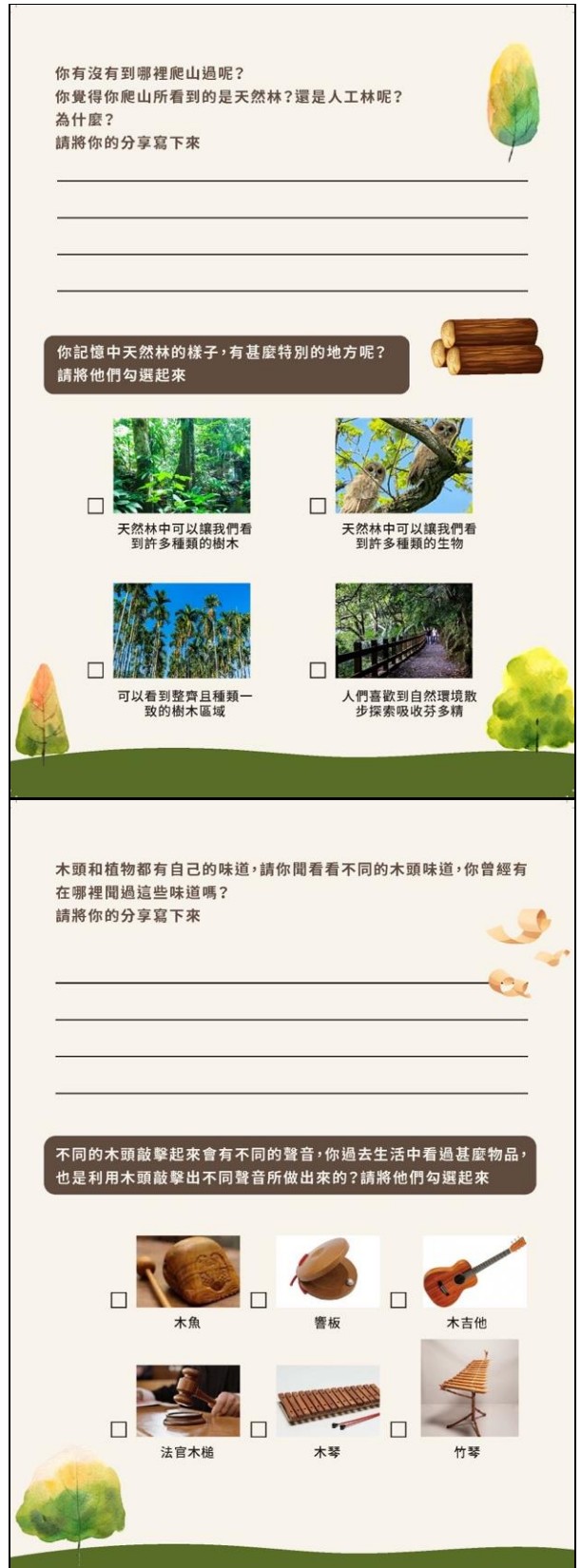

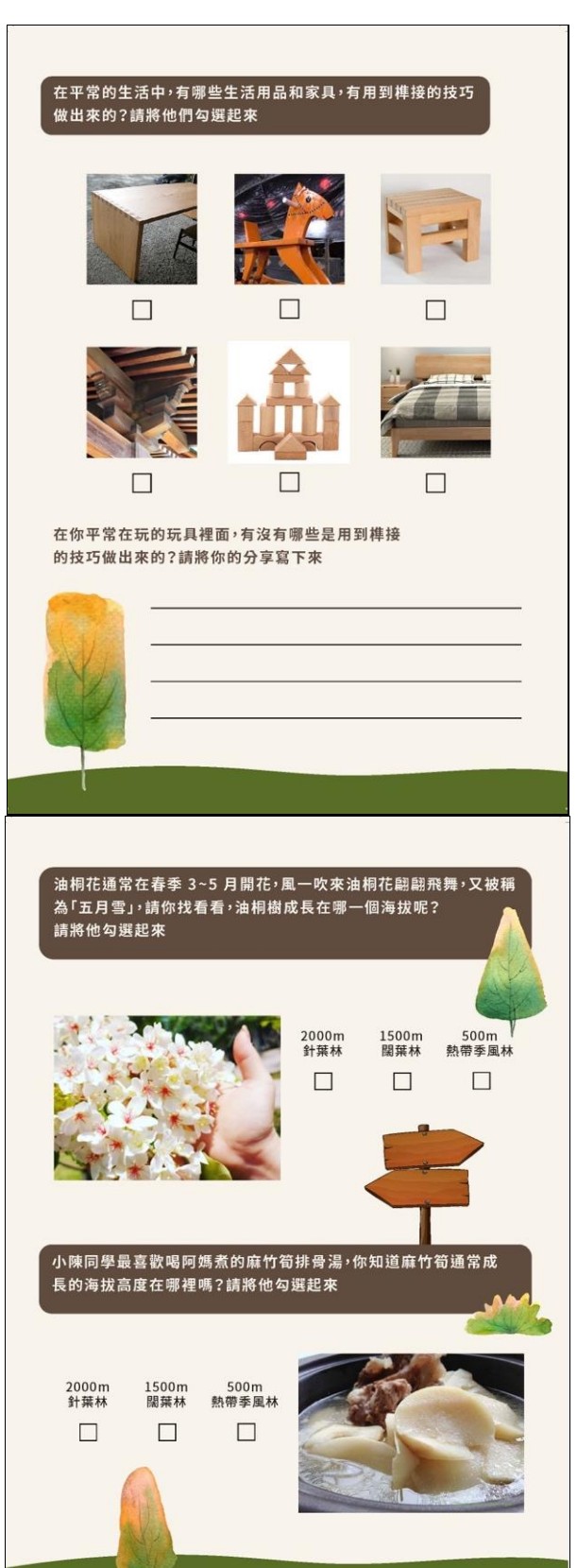

請你回想看看
在三樓和四樓的生活物品中,哪一些通常會用「聞起來有香味」的木頭呢?請你將物品的名字寫下來

_______________________________

_______________________________

_______________________________

請你回想看看
在三樓和四樓的生活物品中,有沒有哪一些,是你以前沒見過,來到這個展覽才看到的呢?請你將物品的名字寫下來

_______________________________

_______________________________

_______________________________

_______________________________

在生活中你有沒有曾經在哪裡看過 CAS 的認證標章?
你曾經在哪裡看到?請你將分享寫下來(至少兩項)

_______________________________

_______________________________

_______________________________

_______________________________

你是否觀察過家具的木頭是否有甚麼不同呢?
一般可以分為實木、夾心版、木心板、密集板、塑合版
其中夾心版、木心板、密集板、合成板多會使用膠合方式製成,請問在你的家中是否也有這樣類型的家具呢?請你將分享寫下來

_______________________________

_______________________________

_______________________________

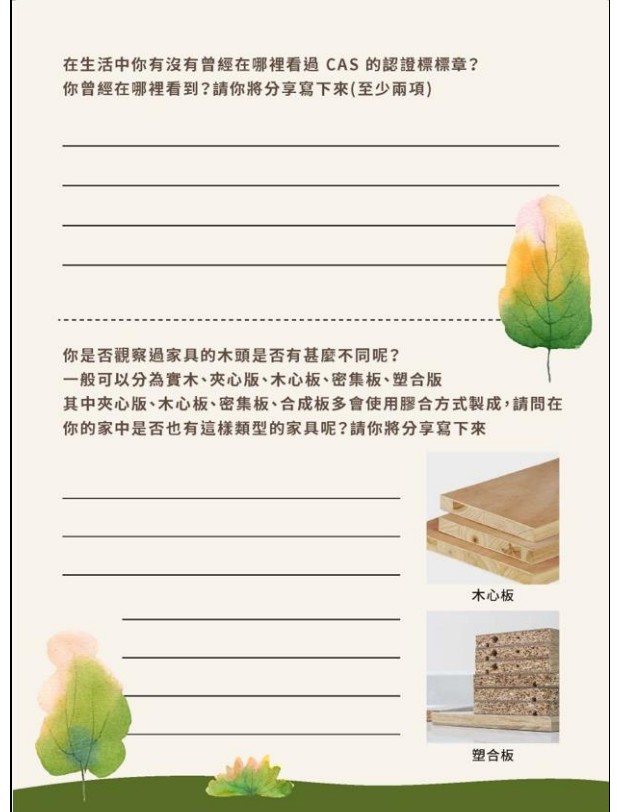

木心板

塑合板

你和你的家人,有沒有自己動手做過木工呢?
是做出了甚麼樣的東西?
你在其中幫忙了甚麼事情呢?請你將分享寫下來

_______________________________

_______________________________

_______________________________

_______________________________

在下列的木工具中,有哪一些工具是你家中有的? 請在 □ 中打叉 ☒
哪一些工具是家裡沒有的呢? 請在 □ 中打叉 ☒

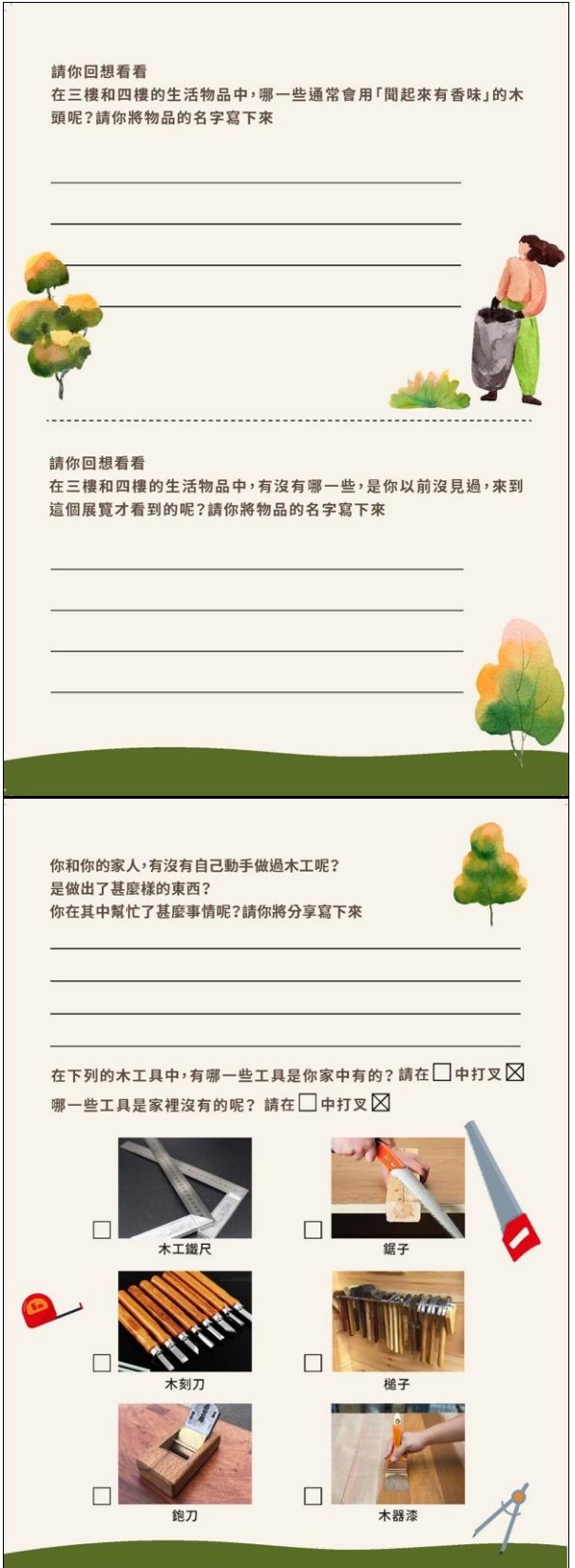

□ 木工鐵尺　　□ 鋸子

□ 木刻刀　　□ 槌子

□ 鉋刀　　□ 木器漆

## Appendix D. "Design Approach of Discussion, Analyzation, Integrated Comparison, and Commentary" (Group D)

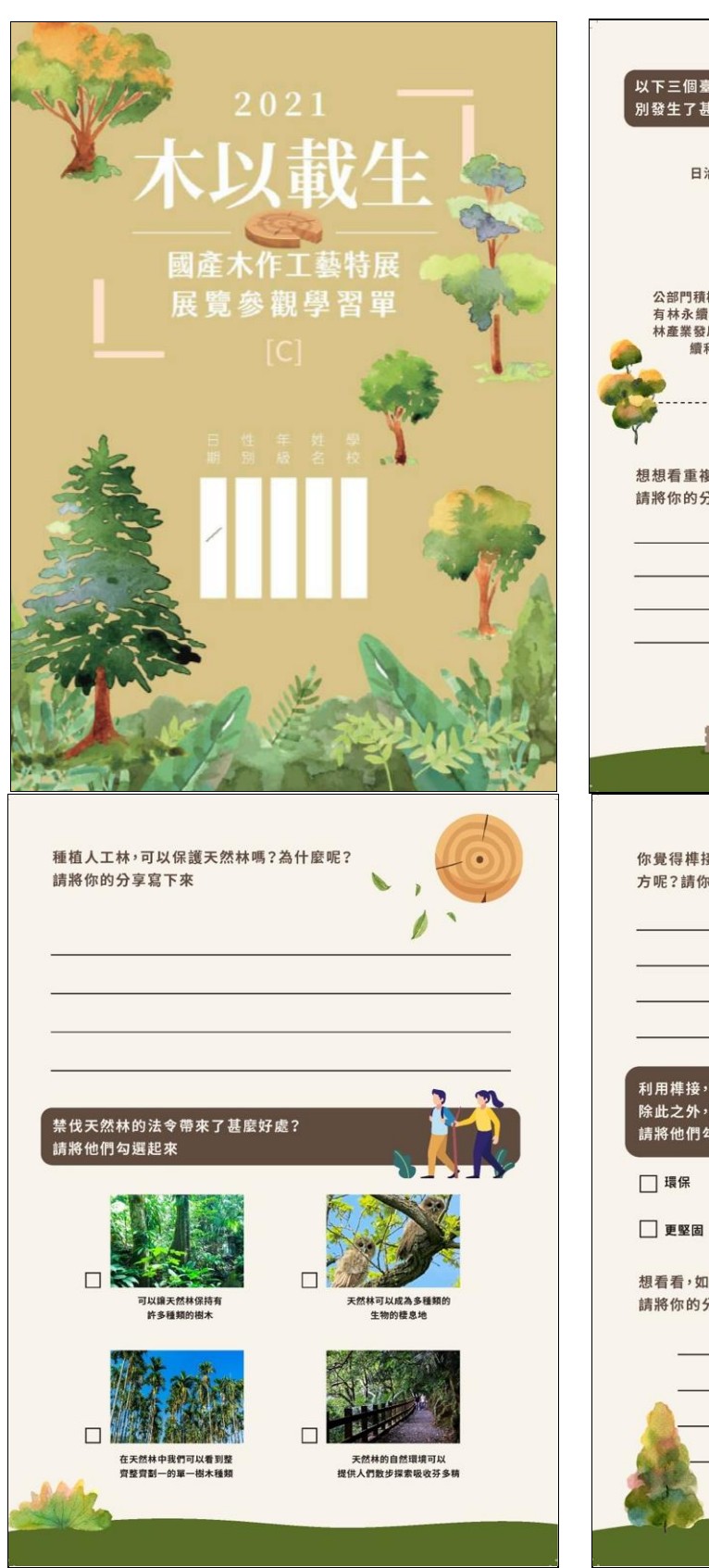

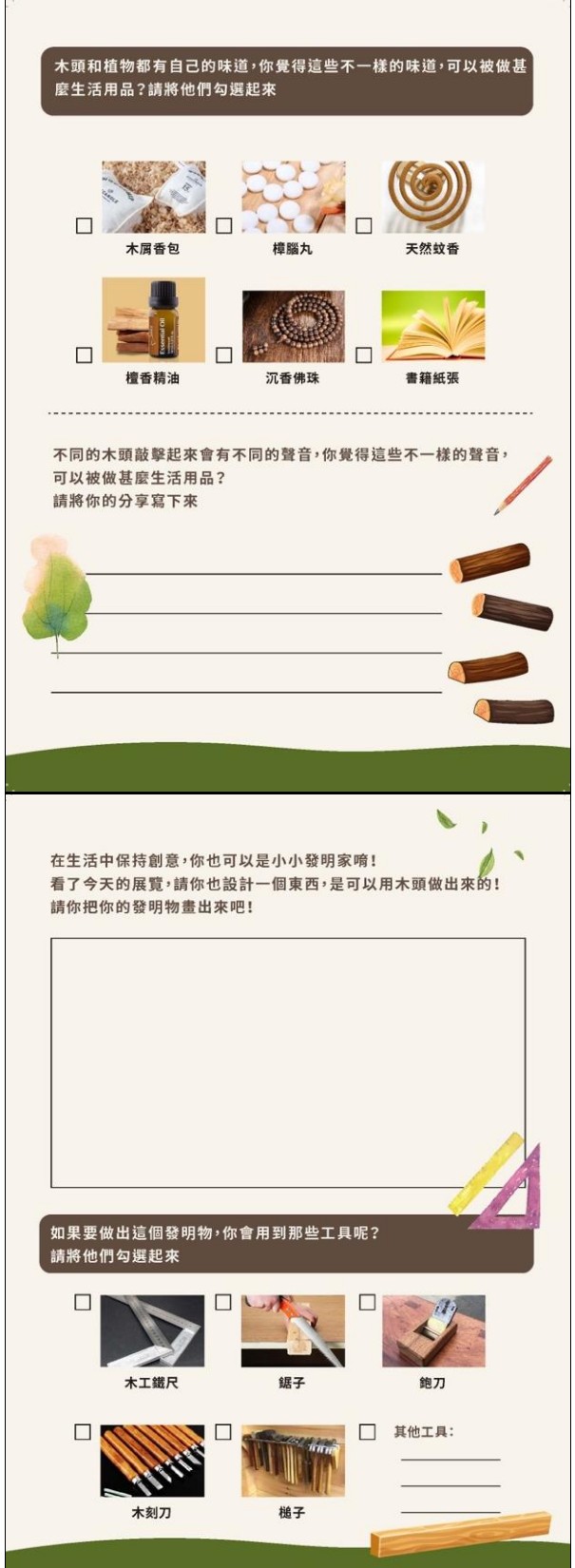

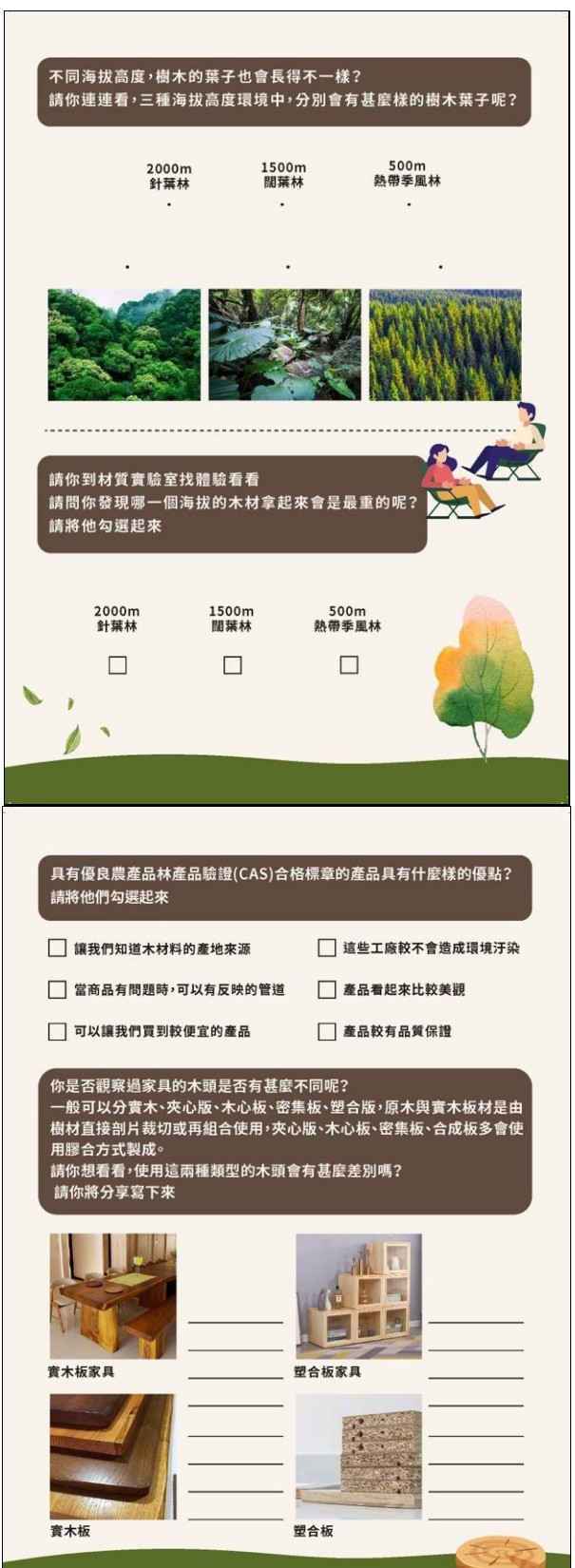

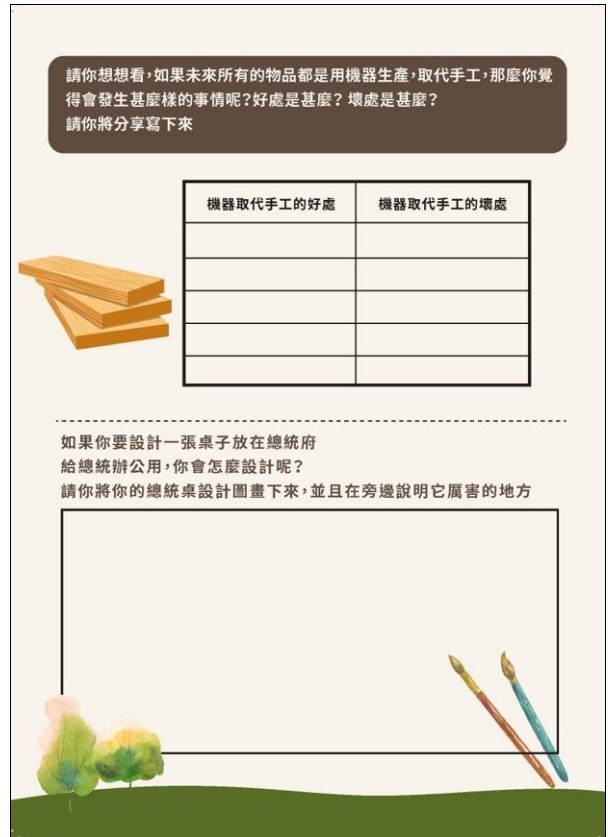

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
