# Peer review of "A Study on the Effects of Digital Learning Sheet Design Strategy on the Learning Motivation and Learning Outcomes of Museum Exhibition Visitors"

_education, doi:10.3390/educsci12020135_

Round 1

Reviewer 1 Report

This study aims to investigate the effects of different design of digital learning sheet on students' motivation and learning outcomes. The study results show that the digital learning sheet designed with the "principle of sensory exploration of physical objects" had better learning motivations and cognitive and psychomotor aspects of learning outcomes. To sum up, it’s an interesting and concise work with contribution. This paper is well-organized and suitable methodologies including ANOVA and the Scheffé test were used for data analysis. The content of this study is also relevant to this journal. Some comments and minor suggestions were provided before publication.

  1. Please present the pre-test and post-test questionnaires, if the questions are the same, just provide one.
  2. Please present the study sheet used by groups B, C, and D.
  3. Tables 1 to 16 are the details of the statistical analysis, and it is not necessary to present them one by one.
  4. It is recommended to aggregate the mean of the post-test results of groups A, B, C, and D into a table, and label the grouping results.
  5. Please discuss the reason of learning sheet designed with the "principle of sensory exploration of physical objects" had better performance than other design sheets. 

Author Response

Comments

Respond

Please present the pre-test and post-test questionnaires, if the questions are the same, just provide one.

The pre-test and post-test questionnaires implemented in this study are attached at the end of this study in Appendix 1.

Please present the study sheet used by groups B, C, and D.

The pre-test and post-test questionnaires implemented in this study are attached in Appendix 2, Appendix 3, and Appendix 4 at the end of this study.

However, doing so will add many pages, so we ask the reviewers to consider whether it is suitable to do so.

Tables 1 to 16 are the details of the statistical analysis, and it is not necessary to present them one by one.

In the revised version, we have reduced the number of prominence tables, including the original Tables 4, 6, 8, 10, 12, 14, and 15, and presented the prominence in the text, leaving only some tables on Scheffé's post-checking.

It is recommended to aggregate the mean of the post-test results of groups A, B, C, and D into a table, and label the grouping results.

In the revised version, we combined Tables 5, 7, 9, and 11 from the original research hypothesis 4 into one table as Table 4, and Tables 13 and 16 from the original research hypothesis 5 into one table as Table 5.

Please discuss the reason of learning sheet designed with the "principle of sensory exploration of physical objects" had better performance than other design sheets.

Thank you for your valuable suggestions!

In the revised version, we have added to the conclusion a discussion of how digital learning sheets designed with the "Sensory and Exploration Design Approach" contribute to better learning outcomes.

Reviewer 2 Report

Thank you for the opportunity to review this body of research. It is well presented and argued. The findings make a valuable contribution to the field of museum education. Suggested changes include:

Consider adding more key words to improve search engine hits - this article deserves to be read!

Editing. e.g. edit sentences (to clarify meaning, remove repetition); some terminology needs better context (espec. 4.2.1 so that a non-statistician understands why 0.000 is relevant).

RE: literature section, remove sentence from line 298-302 and move to section 3. It can serve as an introduction to the Study Methodology, insert at line 349.

Some style things (e.g. some terms written in both upper & lower case).

Conclusions, I think it is good to think about your research outcomes. For example, you can suggest that the next step is that researchers look at studies that use both qualitative and quantitative methods (e.g. survey literature review + fieldwork). Having reviewed your work, I see value in transnational museum education research teams engaging in a larger, collaborative projects.... this would involve meta data & be an interesting idea for Education Science readers to consider.

All suggested changes are noted in the attached file.

Author Response

Respond Review Comments(Reviewer 2)

Review Comments

Respond

Thank you for the opportunity to review this body of research. It is well presented and argued. The findings make a valuable contribution to the field of museum education.

We thank you for the recognition and support of our study. We share the view that this study has an innovative potential, and we look forward to share the results of this study with people in various fields in a high quality international journal.

Consider adding more key words to improve search engine hits - this article deserves to be read!

Thanks to the reviewers for the reminder

In the revised version, we have added some keywords

Editing. e.g. edit sentences (to clarify meaning, remove repetition); some terminology needs better context (espec. 4.2.1 so that a non-statistician understands why 0.000 is relevant).

Thank you for giving us invaluable suggestions to make our study better.

These suggestions have been revised respectively in the latest version.

RE: literature section, remove sentence from line 298-302 and move to section 3. It can serve as an introduction to the Study Methodology, insert at line 349.

We thank you for your suggestions. In the new version, we have adjusted the sentences to make them more readable.

Some style things (e.g. some terms written in both upper & lower case).

Thank you to the reviewers for giving all the suggestions to improve our study.

These suggestions have been revised accordingly in the latest version.

Conclusions, I think it is good to think about your research outcomes. For example, you can suggest that the next step is that researchers look at studies that use both qualitative and quantitative methods (e.g. survey literature review + fieldwork). Having reviewed your work, I see value in transnational museum education research teams engaging in a larger, collaborative projects.... this would involve meta data & be an interesting idea for Education Science readers to consider.

We thank the committee members for acknowledging and supporting our study and giving us invaluable suggestions.

In this new version, we have strengthened the discussion of our conclusions and provided suggestions for future research.
